# Global overview of suicidal behavior and associated risk factors among people living with human immunodeficiency virus: A scoping review

**Yi-Tseng Tsai** [1,2], **Sriyani Padmalatha K. M.** [2,3], **Han-Chang Ku** [1,2], **Yi-Lin Wu** [2], **Nai-Ying Ko** [2,4]*

1 Department of Nursing, An Nan Hospital, China Medical University, Tainan, Taiwan, 2 Department of Nursing, College of Medicine, National Cheng Kung University, Tainan, Taiwan, 3 Operating Room Department, National Hospital of Sri Lanka, Colombo, Sri Lanka, 4 Department of Public Health, College of Medicine, National Cheng Kung University, Tainan, Taiwan

* nyko@mail.ncku.edu.tw

## Abstract

Death by suicide is a major public health problem. People living with human immunodeficiency virus (PLHIV) have higher risk of suicidal behavior than the general population. The aim of this review is to summarize suicidal behavior, associated risk factors, and risk populations among PLHIV. Research studies in six databases from January 1, 1988, to July 8, 2021, were searched using keywords that included "HIV," "suicide," and "risk factors." The study design, suicide measurement techniques, risk factors, and study findings were extracted. A total of 193 studies were included. We found that the Americas, Europe, and Asia have the highest rates of suicidal behavior. Suicide risk factors include demographic factors, mental illness, and physiological, psychological, and social support. Depression is the most common risk factor for PLHIV, with suicidal ideation and attempt risk. Drug overdosage is the main cause of suicide death. In conclusion, the current study found that PLHIV had experienced a high level of suicidal status. This review provides an overview of suicidal behavior and its risk factors in PLHIV with the goal of better managing these factors and thus preventing death due to suicide.

## Introduction

Death due to suicide is a major public health problem worldwide. According to the World Health Organization (WHO), approximately 700,000 people died worldwide due to suicide every year (an average of one death every 40 s) [1]. Suicide is a global phenomenon and can occur at any age. Acquired immunodeficiency syndrome (HIV) and human immunodeficiency virus (AIDS) is also a common public health issue, and currently there are more than 37.9 million people living with HIV/AIDS around the world [2]. The rate of suicide deaths in People living with HIV (PLHIV) is 100-fold higher than the rate that has been reported in the general population [3]. Prevalence estimates of suicidal ideation, attempts, and plans among

**Data Availability Statement:** we analysis data from published article therefore no data availability.

**Funding:** funding for this publication covered by An-Nan Hospital, China Medical University, Tainan,

Taiwan (Grant No: ANHRF109-24). However, the funders had no role in study design, data collection and analysis, decision to publish, or preparation of the manuscript.

**Competing interests:** The authors have declared that no competing interests exist.

people living with HIV/AIDS were more common and serious than those in the general population [4]. Suicide attempt rates among PLHIV with mental disorders and psychiatric treatment have continued to increase from the pre-highly active antiretroviral therapy (Pre-HAART) era (1988–1995) to the HAART era (1996–2008) from 27.8% to 35.1%, respectively [5].

Suicidal behavior is complex, with different levels of severity, ranging from suicidal ideation to suicide attempts and ultimately to the end of life by death due to suicide. Suicidal ideation is defined as thoughts, considerations, or plans to die by suicide, whereas suicide attempts are defined as failed attempts to die by suicide where the person survives [6–8]. That suicidal ideation is more common than suicide attempt and death by suicide, and the presence of suicidal ideation increases the risk of suicide attempt and death by suicide. The suggest a complex interrelationship between behavior and suicide attempts [8, 9]. Suicidal ideation is an important predictor of subsequent suicide attempts and dying by suicide [9, 10].

Suicidal behavior is a complicated process that ranges in degree of severity, from thinking about killing oneself (i.e., suicidal ideation) to doing it (i.e., suicide attempt and death by suicide). In the current study provided insights into the relationships among HARRT, depression, and suicidal status in PLHIV and evidence that depression played a mediating role in the association between suicide ideation and attempt. However, the relationship between these three-suicide behavior is unclear; for example, relationship between HARRT, and death by suicide or depression, and suicide attempts, therefore, this study will be a better feasibility to understanding relationship between these three-suicide behavior and could help prevent suicidal behavior in PLHIV, in whom suicide is a significant public health problem of HIV-infected adults. It is important to categorization of suicidal behaviors among PLHIV due to lack of overview of scope reviewing in this population, even suicide became significant life-threatening event in PLHIV [9].

In general, primary research studies consider only one or two suicidal behavior, such as only suicidal ideation or attempts, or both suicidal ideation and suicide attempts, within a single center or country. Some studies included specific at-risk populations like perinatal women, homosexual men, and prisoners with HIV. However, these studies did not involve all at-risk populations and their risk behavior [11–16]. In previous primary research, among PLHIV, poor social support, HIV stigma, mental disorders, and associated comorbidities were associated with increased suicide rates. Improvements in antiretroviral therapy have led to better survival rates in PLHIV; however, suicidal behavior remain a major health issue [17, 18].

Thus, exploring suicidal behavior using a wide range of global research studies is vital for primary healthcare professionals to plan early recognition of this and suicide prevention strategies. The aim of this review is to provide an overview of the rates of suicidal behavior and associated suicide risk factors among PLHIV. Detection of suicidal ideation and suicide attempts is important in planning early suicide prevention and optimizing HIV/AIDS management.

## Materials and methods

This review has been registered in the International Platform of Registered Systematic Review and Meta-analysis Protocols (INPLASY, Reg No: INPLASY202170033).

### Search strategy

Literature was searched using the following six databases: Embase, Ovid MEDLINE, CEN-TRAL, Web of Science, Academic Search Complete, and Psychology & Behavioral Sciences Collection. This was done after meeting with a public health librarian (the author CJF) and

two members of the research team to clarify goals and further define the selection criteria to develop the literature search strategy. This review included studies published between January 1, 1988 and July 8, 2021 based on the Preferred Reporting Items for Systematic Reviews and Meta-Analyses guidelines [19].

English synonyms such as AIDS, T-lymphotropic virus, or human T-cell lymphotropic virus, type III human T-cell leukemia virus, type III lymphadenopathy-associated virus, LAV-HTLV-III, HTLV-III-LAV, type III infection, or HTLV-III infection were used in each database to identify suicidal behavior among PLHIV. We also used several control phrases from the Emtree and Medical Subject Headings (MeSH) databases. For Emtree, these included "Human immunodeficiency virus," "Human immunodeficiency virus infection," "suicidal behavior," or "automutilation," and "suicide," and for MeSH, they included "HIV infections," "HIV," or "self-injurious behavior." We supplemented the search results with the EndNote X9 bibliographical database, and the search results were manually screened, including the reference lists of relevant articles and previous systematic reviews to confirm the sensitivity of the search strategy (S1 File) [20].

## Eligibility criteria

The inclusion criteria were as follows: (1) the studies provided primary data on the prevalence or incidence of suicidal ideation, suicide attempts, or suicides measured using validated assessment tools or coded medical report data within a population-based study; (2) the participants were aged ≥15 years; (3) the participants were diagnosed with HIV/AIDS; and (4) the report was an original, published article in English or Chinese. The following types of studies were excluded: (1) the study population did not include PLHIV; (2) those unrelated to suicide; and (3) case report and review studies.

Titles and abstracts were independently screened by three researchers based on the inclusion and exclusion criteria after automatically removing duplicates using EndNote X9. Then, the full text of the selected studies was reviewed independently by three researchers, with any disagreement resolved by a fourth researcher to avoid selection bias. Disagreements regarding article inclusion were resolved by discussion between all authors.

## Quality assessment

All eligible studies were assessed for quality of evidence using the Joanna Briggs Institute Critical Appraisal Checklist for Prevalence Studies Scale, which contains nine items and four responses (yes, no, unclear, and not applicable) [21]. Studies with a total score of 8 and above were considered to have high quality evidence and were included in this systematic review. Study quality and risk of bias were independently assessed by three researchers, with any disagreements resolved by a fourth researcher.

## Data extraction

The authors YT and HC conceptualized the study and developed the research protocol. YT created the initial draft of the data extraction chart containing the study characteristics of interest *a priori*. YT, HC, and YL identified articles for full-text review. Full data extraction was then carried out independently on each article by YT, HC, and YL, and any disagreements were resolved through discussion with a fourth author. Data extraction was recorded on a standardized Excel sheet. The following details were listed: the name of the authors and the publication year, the name of the journal, country, setting, study design, sample size, included risk factors for suicide, available measurement tools, and prevalence of suicide ideation, suicide attempts, and completed suicide (Table 1).

**Table 1. Study characteristic of selected 193 studies.**

| Author, Year of publication | Journal | Type of study | WHO region | Country | Sample size | Study setting | Measurement tool | Suicide ideation rate | Suicide attempt rate | Completed suicide rate | Risk factor |
|---|---|---|---|---|---|---|---|---|---|---|---|
| Marzuk, P. M. et al. (1988) [23] | Journal of the American Medical Association | retrospective study | Region of the Americas | United States | 3,828 | Database | N/A | N/A | N/A | 680.56 per 100,000 person-years. | Age 25–59. |
| Perry, S. et al. (1990) [24] | Journal of the American Medical Association | cross-sectional study | Region of the Americas | United States | 301 | Hospital | Beck Depression Inventory (BDI). | 28.6% (n = 86) | N/A | N/A | Depression. |
| Cote, T. R. et al. (1992) [25] | Journal of the American Medical Association | cohort study | Region of the Americas | United States | 98,473 person-years. | Database | N/A | N/A | N/A | 167 per 100,000 person-years. | Drug poisoning, firearms, suffocation, jumping from high places, age, race. |
| Gala, C. et al. (1992) [119] | Acta Psychiatrica Scandinavica | cross-sectional study | European region | Italy | 213 | Hospital | N/A | N/A | N/A | N/A | DSH, psychiatric history. |
| McKegney, F. P. et al. (1992) [26] | American Journal of Psychiatry | cross-sectional study | Region of the Americas | United States | 404 | Hospital | N/A | N/A | N/A | N/A | Organic mental disorders. |
| Rajs, J. et al. (1992) [77] | Acta Psychiatrica Scandinavica | retrospective study | European region | Sweden | 85 | Hospital | N/A | N/A | N/A | 25% (n = 21) | Homo- and bisexual males, intravenous drug addicts, medicinal drug overdosage, low psychosocial support. |
| Brown, G. R. et al. (1993) [27] | Vaccine | longitudinal study | Region of the Americas | United States | 394 | Hospital | Standard anxiety (HARS) and depression (HDRS) rating scale scores. | 17% (n = 67) | N/A | N/A | Anxiety, depression, major mood disorder, psychoactive substance use disorder, medication overdose, cutting wrists, firearms and jumping, violent. |
| Chu, S. Y. et al. (1993) [195] | American Journal of Public Health | retrospective study | Region of the Americas | United States | 19,564 | Database | N/A | N/A | N/A | 0.3% (n = 58) | HIV exposure category and time between diagnosis and death. |
| Twiname, B. G. (1993) [187] | Journal of the Association of Nurses in AIDS Care | cross-sectional study | Region of the Americas | United States | 80 | Hospital | N/A | N/A | N/A | N/A | Depression |
| Alfonso, C. A. et al. (1994) [28] | Psychosomatics | retrospective study | Region of the Americas | United States | 2,363 | Hospital | N/A | N/A | 21.8% (n = 515) | N/A | HIV seropositivity. |
| Craven, D. E. Et al. (1994) [160] | Annals of Internal Medicine | retrospective study | Region of the Americas | United States | 7 | Hospital | N/A | N/A | 42.8% (n = 3) | N/A | CD4 cell count, depression, history of illicit narcotic use. |
| Hanvelt, R. A. et al. (1994) [122] | AIDS | cohort study | Region of the Americas | Canada | 144,876 | Database | N/A | N/A | N/A | 9.4% (n = 13,618) | Indirect cost of production lost |
| Van Haastrecht, H. J. A. Et al. (1994) [29] | AIDS | cohort study | European region | Netherlands | 86 | Hospital | N/A | N/A | N/A | 8.13% (n = 7) | Overdose, intravenous drug-using (IDU) |
| Carvajal, M. J. et al. (1995) [78] | AIDS Care | retrospective study | European region | Spain | 422 | Hospital | N/A | 1.18% (n = 5) | 4.02% (n = 17) | 0.47% (n = 2) | Accidental overdose, IDU. |
| Mancoske, R. J. et al. (1995) [30] | Social Work | cohort study | Region of the Americas | United States | 51 | Database | N/A | N/A | N/A | 175 per 10,000-person year | Nonmetropolitan areas. |

(Continued)

**Table 1.** (Continued)

| Author, Year of publication | Journal | Type of study | WHO region | Country | Sample size | Study setting | Measurement tool | Suicide ideation rate | Suicide attempt rate | Completed suicide rate | Risk factor |
|---|---|---|---|---|---|---|---|---|---|---|---|
| Sherr, L. (1995) [79] | AIDS Care | retrospective study | European region | United Kingdom | 188 | Clinic | N/A | 50.5% (n = 95) | 21.4% (n = 40) | 0.5% (n = 1) | Overdosing, diagnosis with peaks at or around diagnosis. |
| Bindels, P. J. et al. (1996) [31] | Lancet | Retrospective study | European region | Netherlands | 131 | Hospital | N/A | N/A | N/A | 13% (n = 17) | End stage illness. |
| Breitbart, W. et al. (1996) [188] | American Journal of Psychiatry | cross-sectional study | Region of the Americas | United States | 378 | Database | N/A | 55% (n = 207) | N/A | N/A | Euthanasia, physician-assisted suicide (PAS). |
| Dannenberg, A. L. Et al. (1996) [32] | Journal of the American Medical Association | cohort study | Region of the Americas | United States | 4,147 | Database | N/A | N/A | N/A | 49 per 100000 person-years. | Physician- assisted suicide (PAS), depression, hopelessness, psychological distress, social factors. |
| Marzuk, P. M et al. (1997) [33] | American Journal of Psychiatry | cross-sectional study | Region of the Americas | United States | 1,511 | Database | N/A | N/A | N/A | 8.67% (n = 131) | Male, time from screening to death was less than 3 months. |
| Rosengard, C. et al. (1997) [34] | AIDS Care | cross-sectional study | Region of the Americas | United States | 86 | Database | N/A | 12.78% (n = 11) | N/A | N/A | Bereavement, feeling burdened, low social support, subjective social integration. |
| Sherr, L. et al. (1997) [80] | Genitourinary Medicine | retrospective study | European region | United Kingdom | 100 | Clinic | N/A | 69% (n = 69) | 31% (n = 31) | N/A | Female. |
| Wood, K. A. et al. (1997) [35] | AIDS Care | cross-sectional study | Region of theAmericas | United States | 50 | Hospital | N/A | 60% (n = 30) | N/A | N/A | Schizophrenia. |
| Gil, F. et al. (1998) [36] | AIDS Patient Care STDS | cross-sectional study | Region of the Americas | United States | 91 | Hospital | Suicidal ideation (Scale for Suicide Ideation Self-Report). | 63.4% (n = 57) | N/A | N/A | Psychological symptoms, physical symptoms, social support, satisfaction with the social support received, poor sexual adjustment. |
| Kelly, B. et al. (1998) [37] | Psychosomatics | cross-sectional study | Region of the Americas | United States | 164 | Hospital | Beck Depression Inventory and the General Health Questionnaire (28-item version). | N/A | 21% (n = 34) | N/A | Male, psychiatric disorder, neuroticism, unemployment. |
| Swartz, H. A. et al. (1998) [38] | Psychiatric Services | cross-sectional study | Region of the Americas | United States | 33 | Hospital | 1. The 24-item Hamilton Rating Scale for Depression (Ham-D), Beck Depression Inventory (BDI). | N/A | 52% (n = 17) | N/A | Substance abuse. |
| Kalichman, S. C. et al. (2000) [39] | Psychiatric Services | cross-sectional study | Region of the Americas | United States | 113 | Clinic | Beck Depression Inventory (BDI) | 26% (n = 29) | N/A | N/A | Men, whites, gay. |
| Malbergier, A. et al. (2001) [194] | AIDS Care | cross-sectional study | Western Pacific region | Australia | 30 | Hospital | N/A | N/A | 27% (n=8) | N/A | Depression. |
| Bonnet, F. et al. (2002) [81] | HIV Medicine | retrospective study | European region | France | 107 | Hospital | N/A | N/A | N/A | 6% (n = 7) | |

(*Continued*)

**Table 1.** (Continued)

| Author, Year of publication | Journal | Type of study | WHO region | Country | Sample size | Study setting | Measurement tool | Suicide ideation rate | Suicide attempt rate | Completed suicide rate | Risk factor |
|---|---|---|---|---|---|---|---|---|---|---|---|
| Cohen, M. H. et al. (2002) [40] | American Journal of Medicine | cohort study | Region of the Americas | Brazil | 1,902 | Database | N/A | N/A | N/A | (n = 10) | Women, depression, IDU with hepatitis C infection, smoking, age. |
| Heckman, T. G. Et al. (2002) [41] | Ann Behav Med | cross-sectional study | Region of the Americas | United States | 201 | Clinic | N/A | 38% (n = 76) | N/A | N/A | Depression, less coping self-efficacy, transmitting their HIV infection to others, stress associated with AIDS-related stigma. |
| Lochet, P. et al. (2003) [82] | HIV Medicine | cross-sectional study | European region | France | 174 | Hospital | General Health question (GHD–28). | 13.2% (n = 23) | N/A | N/A | N/A |
| Roy, A. (2003) [42] | Acta Psychiatrica Scandinavica | cross-sectional study | Region of the Americas | United States | 149 | Clinic | The depression section of the Structured Clinical Interview for DSM-IV. | N/A | 44.3% (n = 66) | N/A | Female, younger, substance dependence, depression. |
| May, T. et al. (2003) [83] | Presse Medicale | retrospective study | European region | France | 864 | Database | N/A | N/A | N/A | 11% (n = 6) | N/A |
| Summers, J. et al. (2004) [43] | Death Studies | cross-sectional study | Region of the Americas | United States | 93 | Clinic | 1. Hamilton Depression Rating Scale. 2. Suicide Assessment Questions from the National Institute of Mental Health Diagnostic Interview Schedule Version III-A (DIS). | 51.61% (n = 48) | 25.8% (n = 24) | N/A | N/A |
| Cooperman, N. A. et al. (2005) [15] | Journal of Behavioral Medicine | cross-sectional study | Region of the Americas | United States | 207 | Clinic | Suicidal Ideation and Behavior Suicidal ideation was assessed with a scale. | N/A | 26% (n = 54) | N/A | Women, having children, employed. |
| Gielen, A. C. Et al. (2005) [44] | Womens Health Issues | cross-sectional study | Region of the Americas | United States | 310 | Clinic | Intimate partner violence. IPV screening tool widely used in health care settings. | 31% (n = 96) | 16% (n = 49) | N/A | Abused women, anxiety, depression. |
| Krentz, H. B. Et al. (2005) [15] | HIV Medicine | retrospective study | Region of the Americas | Canada | 560 | Hospital | N/A | N/A | N/A | 7% (n = 40) | N/A |
| Lewden, C. et al. (2005) [84] | International Journal of Epidemiology | Retrospective study | Region of the Americas | United States | 964 | Hospital | N/A | N/A | N/A | 4% (n = 38) | HIV infection had been diagnosed recently, smoking, alcohol. |
| Olley, B. O. et al. (2005) [123] | AIDS Care | cross-sectional study | African region | South Africa | 149 | Hospital | The MINI International Neuropsychiatric Interview (MINI). | N/A | 54% (n = 12) | N/A | Posttraumatic stress disorder. |
| Petrushkin, H. et al. (2005) [124] | Psychiatric Bulletin | cross-sectional study | African region | Uganda | 46 | Hospital | The MINI International Neuropsychiatric Interview (MINI). | N/A | 17.39% (n = 8) | N/A | Depression. |
| Jin, H. et al. (2006) [85] | Journal of Affective Disorders | cross-sectional study | European region | France | 28 | Hospital | Beck Depression Inventory-I (BDI). | 18% (n = 4) | N/A | N/A | Depression. |
| Lu, T. H. et al. (2006) [161] | Journal of the Formosan Medical Association | cohort study | Western Pacific Region | China | 752 | Hospital | The MINI International Neuropsychiatric Interview (MINI). | N/A | N/A | 4.8% (n = 14) | HAART. |

(*Continued*)

**Table 1.** (Continued)

| Author, Year of publication | Journal | Type of study | WHO region | Country | Sample size | Study setting | Measurement tool | Suicide ideation rate | Suicide attempt rate | Completed suicide rate | Risk factor |
|---|---|---|---|---|---|---|---|---|---|---|---|
| Olley, B. O. (2006) [125] | African Journal of AIDS Research | cross-sectional study | African region | South Africa | 105 | Hospital | The MINI International Neuropsychiatric Interview (MINI). | 11.4% (n = 12) | N/A | N/A | Depression. |
| Robertson, K. et al. (2006) [162] | Death Stud | cross-sectional study | Western Pacific Region | Taiwan | 191 | Hospital | The MMPI-2 is a well standardized self-report instrument consisting of 13 basic scales. | 39.27% (n = 75) | 26.70% (n = 51) | N/A | N/A |
| Shelton, A. J. et al. (2006) [45] | AIDS Care | cross-sectional study | Region of the Americas | United States | 54 | Clinic | 1. A semi-structured interview; Brief Symptom Inventory (BSI; Derogatis & Spencer, 1982). 2. The MMPI-2 is a well standardized self-report instrument consisting of 13 basic scales. 3. Ten clinical scales (Hypochondriasis, Depression, Hysteria, Psychopathic Deviate, Masculinity-Femininity, Paranoia, Psychasthenia, Schizophrenia, Hypomania, Social Introversion). | 59.26% (n = 32) | 29.63 (n = 16) | N/A | White. |
| Carrico, A. W. et al. (2007) [46] | Aids | cross-sectional study | Region of the Americas | United States | 2,909 | Clinic | N/A | 19% (n = 553) | N/A | N/A | Marijuana use, depression. |
| Lewden, C. et al. (2008) [86] | Jaids-Journal of Acquired Immune Deficiency Syndromes | cohort study | Region of the Americas | United States | 1,042 | Hospital | 1. Perceived social support the 24-item Social Provisions Scale (Cronbach's a = 0.82). 2. Coping self-efficacy Coping self-efficacy was assessed with an abbreviated (15-item) version of a 26-item scale (Cronbach's a = 0.92). | N/A | N/A | 5% (n = 52) | Male, CD4 cell count. |
| Lifson, A. R. et al. (2008) [87] | HIV Clinical Trials | case-controlled study | Region of the Americas | United States | 11,593 | Hospital | N/A | N/A | N/A | 0.2% (n = 23) | N/A |
| Preau, M. et al. (2008) [120] | AIDS Care | cross-sectional study | European region | France | 2,932 | Hospital | Depression symptoms and suicidal ideation the 21-item Beck Depression Inventory (BDI) (Cronbach's a = 0.86). | N/A | 22% (n = 645) | N/A | Born in France, female, younger adults, lower level of education, unemployed, difficult household financial situation. |

*(Continued)*

**Table 1.** (Continued)

| Author, Year of publication | Journal | Type of study | WHO region | Country | Sample size | Study setting | Measurement tool | Suicide ideation rate | Suicide attempt rate | Completed suicide rate | Risk factor |
|---|---|---|---|---|---|---|---|---|---|---|---|
| Quintana-Ortiz, R. A. Et al. (2008) [121] | Ethn Dis | cohort study | European region | France | 714 | Database | N/A | N/A | 22% (n = 157) | N/A | Male, HIV/AIDS status at study entry, IDU, stress factors related to filial relationships, psychoactive substance, isolation, depression, anxiety. |
| Sherr, L. et al. (2008) [47] | AIDS Care | cross-sectional study | Region of the Americas | United States | 778 | Clinic | N/A | 31% (n = 241) | N/A | N/A | Heterosexual man, black, white, unemployment, lack of disclosure of HIV status, stopped antiretroviral treatment, physical symptoms, psychological symptoms, poorer quality of life. |
| Yang, C. H. et al. (2008) [163] | HIV Medicine | retrospective study | Western Pacific Region | Taiwan | 1,161 | Hospital | N/A | N/A | N/A | 5.5% (n = 62) | |
| Lawler, K. et al. (2009) [126] | AIDS and Behavior | cross-sectional study | African region | Africa | 120 | Hospital | 1. Beck Depression Inventory-Fast Screen for Medical Patients (BDI-FS). 2. Mood Module (MM) of the Primary Care Evaluation of Mental Disorders (Prime-MD) | 12% (n = 15) | N/A | N/A | Depression. |
| Shacham, E. et al. (2009) [48] | AIDS Patient Care & STDs | cross-sectional study | Region of the Americas | United States | 514 | Clinic | The Patient Health Questionnaire (PHQ-9). | 15% (n = 78) | N/A | N/A | unemployed. |
| Hessamfar-Bonarek, M. Et al. (2010) [88] | International Journal of Epidemiology | retrospective study | European region | United Kingdom | 1,013 | Database | N/A | N/A | N/A | women 4% (n = 40) men 9% (n = 91) | Female, poor socio-economic conditions. |
| Keiser, O. et al. (2010) [5] | American Journal of Psychiatry | cohort study | European region | Switzerland | 15,275 | Database | N/A | N/A | N/A | 150 died by suicide (rate 158.4 per 100,000 person-years). | Older patients, male, IDU, patients with advanced clinical stage of HIV illness. |
| Lampe, F. C. et al. (2010) [89] | Journal of Acquired Immune Deficiency Syndromes | cross-sectional study | European region | United Kingdom | 188 | Clinic | N/A | N/A | 27% (n = 51) | N/A | Depression, anxiety. |
| Lau, J. T. et al. (2010) [164] | AIDS Care | cross-sectional study | Western Pacific Region | China | 176 | Clinic | Depression, Anxiety, and Stress Scales (Cronbach's 0.80 to 0.83). | 34% (n = 60) | 8% (n = 14) | N/A | Depression, anxiety, stress. |
| Lawrence, S. T., et al. (2010) [10] | Clinical Infectious Diseases | cohort study | Region of the Americas | United States | 1,216 | Clinic | The psycho-social domains assessed by the PROs include depression (PHQ-9). | 14% (n = 170) | N/A | N/A | Male, white, young middle-aged, substance abuse, depression. |
| Peng, E. Y. et al. (2010) [165] | AIDS Care | cross-sectional study | Western Pacific Region | Taiwan | 535 | Prisons | The five-item Brief Symptom Rating Scale (BSRS-5). | 12.5% (n = 67) | (n = 22) | N/A | Depression, anxiety, psychological distress. |

(*Continued*)

**Table 1.** (Continued)

| Author, Year of publication | Journal | Type of study | WHO region | Country | Sample size | Study setting | Measurement tool | Suicide ideation rate | Suicide attempt rate | Completed suicide rate | Risk factor |
|---|---|---|---|---|---|---|---|---|---|---|---|
| Peng, E. Y. C. Et al. (2010) [166] | Journal of the Formosan Medical Association | cross-sectional study | Western Pacific Region | Taiwan | 479 | Prisons | The five-item Brief Symptom Rating Scale (BSRS-5). | N/A | 4.2% (n = 20) | N/A | Psychiatric morbidity, physical pain or discomfort, depression, anxiety. |
| Schlebusch, L. et al. (2010) [127] | African Journal of Psychiatry | cohort study | African region | South Africa | 112 | Hospital | Comprehensive mental state examination and administration of a semi structured questionnaire to obtain biographical, socio-demographic and other relevant data. | N/A | 67.2 per 100, 000 person-years. | N/A | N/A |
| Aldaz, P. et al. (2011) [90] | Bmc Public Health | cohort study | European region | Spain | 1145 | Database | N/A | N/A | N/A | (n = 7) | IDU. |
| Atkinson, J. H. et al. (2011) [167] | Journal of Affective Disorders | longitudinal study | Western Pacific Region | China | 203 | Clinic | World Mental Health Composite International Diagnostic Interview (WMH-CIDI, version 3.0). | 49.26% (n = 100) | N/A | N/A | Depression. |
| Cejas M, R et al. (2011) [92] | Journal of Psychosomatic Research | cross-sectional study | European region | Romania | 125 | Hospital | 1. Calgary Depression Scale. Plutchick Suicide Risk Scale. | 12.1% (n = 66) | N/A | N/A | Abuse drug, psychiatric disorder, previous suicide attempts, CD4 levels (poor immune status). |
| Davis, S. J. et al. (2011) [49] | Journal of Rehabilitation | cross-sectional study | Region of the Americas | United States | 71 | Clinic | 1. Suicidal/Homicidal Thought Scale. Depression Symptom Scale. | 5.7% (n = 4) | N/A | N/A | N/A |
| Halman, M. et al. (2011) [50] | Canadian Journal of Infectious Diseases and Medical Microbiology | Retrospective study | Region of the Americas | Canada | 87 | Palliative care center | 1. Modified HIV Stressor Scale. 2. Social Support Scale. | 9.2% (n = 8) | 10.3% (n = 9) | N/A | Substance abuse |
| Kinyanda, E. et al. (2011) [128] | BMC Psychiatry | cohort study | African region | Uganda | 618 | Clinic | N/A | N/A | 5.99% (n = 37) | N/A | Female, family history of mental illness, negative coping style, alcohol dependency disorder, food insecurity, Stress. |
| Lee, B. et al. (2011) [168] | Journal of the International Association of Physicians in AIDS Care | case-control study | South-East Asian region | Thailand | 219 | Hospital | N/A | N/A | N/A | 15.5% (n = 34) | Depression. |
| Badiee, J. et al. (2012) [20] | Journal of Affective Disorders | cross-sectional study | Region of the Americas | United States | 1,560 | Clinic | 1. Stress Score index' 5. Social support index' Cronbach of 0.84. 2. M.I.N.I. neuropsychiatric interview (MINI Plus). | 26% (n = 405) | 13% (n = 204) | N/A | Substance abuse, depression, mood disruption. |
| Capron, D.W. et al. (2012) [51] | AIDS Patient Care STDS | cross-sectional study | Region of the Americas | United States | 164 | Database | Children's Depression Inventory (CDI). | N/A | N/A | N/A | Negative affectivity. |
| Chikezie, U. E. et al. (2012) [129] | AIDS Care | cross-sectional study | African region | Nigeria | 150 | Hospital | Beck Depression Inventory-II (BDI-II). | 34.7% (n = 52) | 9.3% (n = 14) | N/A | Older age, female, unemployed, single, no children, living alone, comorbid illness, partner who was also with the disease. |

(*Continued*)

**Table 1.** (Continued)

| Author, Year of publication | Journal | Type of study | WHO region | Country | Sample size | Study setting | Measurement tool | Suicide ideation rate | Suicide attempt rate | Completed suicide rate | Risk factor |
|---|---|---|---|---|---|---|---|---|---|---|---|
| Ellis, J. et al. (2012) [93] | HIV Medicine | retrospective study | European Region | United Kingdom | 46 | Clinic | N/A | N/A | 13% (n = 6) | N/A | Alcohol misuse, recreational drug use, mental health problems. |
| Govender, R. D. et al. (2012) [130] | South African Journal of Psychiatry | cross-sectional study | African region | South Africa | 157 | Hospital | 1. Positive and Negative Affect Scale (PANAS) (range of alpha coefficients: 0.85 to 0.93). 2. Anxiety Sensitivity Index-3 (ASI-3) (Cronbach a = 0.95). | 17.1% (n = 27). | N/A | N/A | Depression. |
| Govender, R. D. et al. (2012) [131] | South African Journal of Psychiatry | cross-sectional study | African region | South Africa | 109 | Hospital | 1. Beck Hopelessness Scale (BHS). 2. Beck Depression Inventory (BDI). | N/A | N/A | N/A | N/A |
| Jia, C. X. et al. (2012) [94] | The Journal of Clinical Psychiatry | cohort study | European Region | Denmark | 9,900 | Database | Beck Depression Inventory (BDI-II). | N/A | N/A | 38.6% (n = 3,821) | Single people, low income, psychiatric illness, first time recently, were treated as inpatients, had a recent hospital contact, had multiple hospital contacts because of the illness. |
| Kinyanda, E. et al. (2012)[132] | BMC Psychiatry | cross-sectional study | African region | Uganda | 618 | Clinic | 1. Beck Hopelessness Scale (BHS). 2. Beck Depression Inventory (BDI). | N/A | 3.9% (n = 24) | N/A | Female, negative life events, previous psychiatric history, major Depression disorder. |
| Lewis, E. L. et al. (2012) [133] | Health Care for Women International | cross-sectional study | Region of the Americas | United States | 62 | Hospital | 1. Beck Depression Inventory-Fast Screen (BDI-FS). 2. Mood Module (MM) of the Primary Care Evaluation of Mental Disorders (Prime-MD). | 11% (n = 7) | N/A | N/A | Female. |
| Protopopescu, C. et al. (2012) [95] | Antiviral Therapy | cohort study | European Region | France | 1,095 | Database | N/A | 1.28 (n = 14) | N/A | 0.46 (n = 5) | N/A |
| Schlebusch, L. et al. (2012) [134] | International Journal of Environmental Research and Public Health | cross-sectional study | African region | South Africa | 189 | Hospital | N/A | 24.2% (n = 46) | N/A | N/A | Young age, male. |
| Sherr, L. et al. (2012) [96] | Women & Health | cross-sectional study | European Region | United Kingdom | 262 | Clinic | Health-Related Quality of Life (HRQOL). | N/A | 34.73% (n = 91) | N/A | Male. |
| Singh, S. et al. (2012) [97] | HIV Medicine | retrospective study | European Region | United Kingdom | 106 | Hospital | N/A | N/A | 6.6% (n = 7) | N/A | N/A |
| Tseng, Z. H. et al. (2012) [52] | Journal of the American College of Cardiology (JACC) | cohort study | Region of the Americas | United States | 2,860 | Clinic | 1. Stress Score index'. 2. Social support index'. | N/A | N/A | 1.54% (n = 44) | Overdoses. |

(*Continued*)

**Table 1.** (Continued)

| Author, Year of publication | Journal | Type of study | WHO region | Country | Sample size | Study setting | Measurement tool | Suicide ideation rate | Suicide attempt rate | Completed suicide rate | Risk factor |
|---|---|---|---|---|---|---|---|---|---|---|---|
| Yaroslavtseva, T. et al. (2012) [98] | European Neuropsychopharmacology | cross-sectional study | European Region | Russia | 700 | Hospital | Beck Depression Inventory (BDI). | 56% (n = 392) | 36% (n = 252) | N/A | Drug dependence, alcohol dependence, HIV stigma, depression. |
| Amiya, R. M. et al. (2013) [169] | Sexually Transmitted Infections | cross-sectional study | South-East Asian region | Nepal | 321 | Clinic | 1. Beck Hopelessness Scale (BHS) 2. Beck Depression Inventory (BDI). | 14% (n = 45) | 17% (n = 54) | N/A | Depression, on ART for more than 2 years. |
| Fernandez-Santos, D. M. Et al. (2013) [53] | Sexually Transmitted Infections | cross-sectional study | Region of the Americas | United States | 499 | Hospital | N/A | N/A | 23.7% (n = 118) | N/A | Smoking, alcohol, psychoactive substances, Intravenous drug usage (IVDU) |
| Jin, H. et al. (2013) [170] | Journal of Acquired Immune Deficiency Syndromes | cross-sectional study | Western Pacific Region | China | 204 | Database | N/A | N/A | 43.1% (n = 88) | N/A | IDU, depression, low social support, stress, alcohol use disorder. |
| Joge, U. S. et al. (2013) [171] | Indian Journal of Dermatology, Venereology & Leprology | cross-sectional study | South-East Asian region | India | 801 | Hospital | N/A | 12.25 (n = 98) | N/A | N/A | Male. |
| Kudryashova H, L. (2013) [54] | Sexually Transmitted Infections | cross-sectional study | Region of the Americas | United States | 350 | Clinic | N/A | N/A | 0.57% (n = 2) | N/A | Substance abuse, mental health. |
| O'Donnell, J. K. et al. (2013) [55] | American Journal of Epidemiology | cross-sectional study | Region of the Americas | United States | 69 | Hospital | Hamilton Rating Scale for Depression. | 28% (n = 19) | N/A | N/A | Depression, psychiatric comorbidities, marital status, adaptive coping styles stress. |
| Schadé, A. et al. (2013) [99] | BMC Psychiatry | cohort study | European Region | Netherlands | 196 | Hospital | Beck Scale for Suicide ideation (BSS). | 48% (n = 94) | 34% (n = 66) | N/A | Depression, fear, anger, guilt. |
| Heuvel, L. V. D. et al. (2013) [135] | AIDS Care | cross-sectional study | African region | Zambia | 649 | Clinic | Mini International Neuropsychiatric Interview (M.I.N.I.). | 5.9% (n = 38) | 3% (n = 19) | N/A | Depression. |
| Weber, R. et al. (2013) [100] | HIV Medicine | cohort study | European Region | Switzerland | 16,134 | Database | N/A | N/A | N/A | 6% (n = 28) | N/A |
| Amiya, R. M. Et al. (2014) [172] | PLoS ONE | cross-sectional study | African region | Nepal | 322 | Clinic | Beck Depression Inventory (BDI). | 14% (n = 45) | N/A | N/A | Depression, lower family support. |
| Ceccon, R. F. et al. (2014) [16] | Revista de saúde pública | cross-sectional study | South-East Asian region | Brazil | 161 | Clinic | 1. Suicide (QIS—Suicide Ideation Questionnaire). 2. Suicide Ideation Questionnaire16 adapted by Ferreira & Castela. | 50% (n = 80) | N/A | N/A | Age at first sexual intercourse < 15 years, children, poverty, living with HIV for long, violence, female. |
| Ehren, K. et al. (2014) [136] | Infection | cohort study | European Region | Germany | 3,165 | Database | N/A | N/A | N/A | 4% (n = 7) | N/A |
| Forbes, K. et al. (2014) [56] | Archives of Disease in Childhood | cross-sectional study | Region of the Americas | United States | 54 | Palliative care center | N/A | N/A | 31.5% (n = 17) | N/A | Sad, hopeless, have had sex against their will, aged 12 or younger at sexual debut, psychosocial support. |

*(Continued)*

**Table 1.** (Continued)

| Author, Year of publication | Journal | Type of study | WHO region | Country | Sample size | Study setting | Measurement tool | Suicide ideation rate | Suicide attempt rate | Completed suicide rate | Risk factor |
|---|---|---|---|---|---|---|---|---|---|---|---|
| Guillemi, S. et al. (2014) [57] | Topics in Antiviral Medicine | cohort study | Region of the Americas | Canada | 5,229 | Hospital | N/A | N/A | N/A | 8.2% (n = 82) | IDU, never having been diagnosed with an AIDS defining illness. |
| Hogg, R. S. Et al. (2014) [16] | Canadian Journal of Infectious Diseases and Medical Microbiology | retrospective study | Region of the Americas | Canada | 5,229 | Hospital | N/A | N/A | N/A | 1998year: 961 deaths per 100, 000 person- years- 2010year: 2.81 deaths per 100, 000 person years. 1998 year: 25%-2010 year: 1.3%. | Younger age, IDU, higher last CD4 count, never having an AIDS defining illness. |
| Jovet-Toledo, G. G. et al. (2014) [58] | AIDS Care | Retrospective study | Region of the Americas | Puerto Rico | 1,185 | Clinic | N/A | N/A | 20.4% (n = 242) | N/A | Gender, employment, drug use, sex work. |
| Kim, et al. (2014) [137] | Journal of the International AIDS Society | cross- sectional study | African region | Malawi | 562 | Clinic | 1. Beck Depression Inventory-II (BDI-II). 2. Children's Depression Inventory-II-Short (CDI-II-S). | 7.1% (n = 40) | N/A | N/A | Depression. |
| McManus, H. et al. (2014) [189] | PLoS ONE | cohort study | European Region | Australia | 81 | Database | N/A | 67% (n = 55) | 85.18% (n = 69) | N/A | MSM, IDU, unemployment, living alone. |
| Mollan, K. R. et al. (2014) [59] | Annals of Internal Medicine | cohort study | Region of the Americas | United States | 5,332 | Hospital | N/A | N/A | 8.08% (n = 62) | N/A | Efavirenz. |
| Passos, S. M. et al. (2014) [17] | AIDS Care | cross- sectional study | Region of the Americas | Brazil | 211 | Clinic | 1. Brazilian version of Hospital Anxiety and Depression Scale (HAD). 2. Module C of the instrument Mini International Neuropsychiatric Interview 5.0 (MINI). | 34.1% (n = 72) | N/A | N/A | Female, age up to 47 years, unemployment, anxiety, depression, abuse or addiction on psychoactive substances. |
| Anagnostopoulos, A. et al. (2015) [101] | PLoS ONE | cohort study | European Region | Switzerland | 4,222 | Hospital | N/A | N/A | N/A | 0.43% (n = 18) | Female, older age, preserved work ability and higher physical activity, depression, IDU. |
| Croxford, S. et al. (2015) [102] | HIV Medicine | cohort study | European Region | United Kingdom | 83,276 | Database | N/A | N/A | N/A | 4.25 per 10000 person-years. | N/A |
| Dabaghzadeh, F. et al (2015) [185] | Iranian Journal of Psychiatry | cross- sectional study | Eastern Mediterranean region | Iran | 150 | Clinic | 1. Hospital Anxiety and Depression Scale (HADS). 2. Positive and Negative Suicide Ideation (PANSI). | Suicidal ideation | N/A | N/A | Anxiety, depression, poor physical activity, sleep quality, unemployment, living alone, lack of family support. |
| Gonzalez-Torres, M. A. et al. (2015) [103] | Neuropsychiatric Disease and Treatment | cross- sectional study | European Region | Spain | 25 | Hospital | N/A | N/A | 72% (n = 18) | N/A | Substance. |

(*Continued*)

**Table 1.** (Continued)

| Author, Year of publication | Journal | Type of study | WHO region | Country | Sample size | Study setting | Measurement tool | Suicide ideation rate | Suicide attempt rate | Completed suicide rate | Risk factor |
|---|---|---|---|---|---|---|---|---|---|---|---|
| Gurm, J. et al. (2015) [60] | CMAJ Open | cohort study | Region of the Americas | Canada | 5,229 | Database | N/A | N/A | N/A | 8.2% (n = 428) | IDU, having no experience with an AIDS-defining illness. |
| Ogundipe, O. A. et al. (2015) [18] | Archives of Suicide Research | cross-sectional study | African region | Nigeria | 295 | Hospital | Beck Depression Inventory (BDI). | N/A | 13.6% (n = 40) | N/A | Poorer quality of life, unemployment, emotional distress, religion, HIV status non-disclosure, previous suicidal attempt. |
| Schlebusch, L. et al. (2015) [138] | Depress Research Treatement | cross-sectional study | African region | South Africa | 157 | Hospital | 1. Beck Hopelessness Scale (BHS). 2. Beck Depression Inventory (BDI). | 28.8% (n = 45) | N/A | N/A | Younger age group (age < 30 years), lower level of education, socioeconomic pressures, traditional beliefs. |
| Wu, Y. L. et al. (2015) [12] | International Journal of STD & AIDS | cross-sectional study | Western Pacific Region | China | 184 | Hospital | 1. Beck Hopelessness Scale (BHS). 2. Beck Depression Inventory (BDI). | 31% (n = 57) | N/A | N/A | Stigma, depression, anxiety. |
| Bitew, H. et al. (2016) [139] | Depress Res Treat | cross-sectional study | African region | Ethiopia | 393 | Hospital | 1. Patients Health Questionnaire version-9 (PHQ 9). 2. Oslo Social Support scale. | 33.6% (n = 132) | 20.1% (n = 79) | N/A | Female, marital status, depression, CD4 level, opportunistic infection, stigma, poor social support. |
| Cheung, C. C. et al. (2016) [61] | HIV Medicine | cohort study | Region of the Americas | Canada | 8,185 | Hospital | N/A | N/A | N/A | 0.47 per 100 person-years. | Age, gender, IDU, AIDS diagnoses, CD4 cell counts. |
| Collins, P. Y. et al. (2016) [140] | Community mental health journal | cross-sectional study | African region | South Africa | 594 | Clinic | Patients Health Questionnaire version-9 (PHQ 9). | Suicidal ideation | N/A | N/A | Psychological distress, Socioeconomic status. |
| Croxford, S. et al. (2016) [104] | HIV Medicine | cohort study | European Region | United Kingdom | 83,276 | Database | N/A | N/A | N/A | 0.23% (n = 190) | N/A |
| de Almeida, S. M. et al. (2016) [62] | Journal of Neurovirology | cross-sectional study | Region of the Americas | Brazil | 39 | Hospital | Beck Depression Inventory-II (BDI-II). | N/A | 18% (n = 7) | N/A | Depression. |
| Fawzi, M. C. S. et al. (2016) [137] | Pediatrics | cross-sectional study | African region | Rwanda | 193 | Clinic | N/A | N/A | 12% (n = 7) | N/A | Conduct problems, depression. |
| Kang, C. R. et al. (2016) [173] | AIDS Care | cross-sectional study | Western Pacific region | South Korea | 422 | Clinic | N/A | 44% (n = 193) | 11%. (n = 47) | N/A | Young and middle age, living with someone, opportunistic disease, depression, lower social support, psychological status. |
| Mugisha, J. et al. (2016) [142] | African Health Sciences | cross-sectional study | African region | Uganda | 2,400 | Clinic | Duong scale (Cronbach's alpha = 0.81). | 12.1%. (n = 290) | 6.2% (n = 149) | N/A | Female, depression, post-traumatic stress disorder. |
| O'Donnell, J. K. et al. (2016) [63] | Journal of Affective Disorders | longitudinal study | Region of the Americas | United States | 289 | Database | M.I.N.I. neuropsychiatric interview (MINI Plus). | 7-19% (n = 21) | N/A | N/A | N/A |

(Continued)

Table 1. (Continued)

| Author, Year of publication | Journal | Type of study | WHO region | Country | Sample size | Study setting | Measurement tool | Suicide ideation rate | Suicide attempt rate | Completed suicide rate | Risk factor |
|---|---|---|---|---|---|---|---|---|---|---|---|
| Rukundo, G. Z. et al. (2016) [143] | Aids Research and Treatment | cross-sectional study | African region | Uganda | 543 | Clinic | 1. The Hamilton Rating Scale for Depression (HAM-D). 2. Mini International Neuropsychiatric Interview (MINI). | 8.8% (n = 48) | 3.1% (n = 17) | N/A | Anger, depression, hopelessness, anxiety, low social support, inability to provide for others, stigma. |
| Rukundo, G. Z. et al. (2016) [144] | African Journal of AIDS Research | cross-sectional study | African region | Uganda | 543 | Clinic | 1. Perceived Social Support Questionnaire (PSSQ). 2. HIV Stigma Scale Questionnaire (HSSQ). 3. Beck Hopelessness Scale (BHS). | 10% (n = 54) | 3%. (n = 17) | N/A | Poor physical health, physical pain, reducing work due to illness, recent HIV diagnosis. |
| Sherr, L. et al. (2016) [105] | Journal of Virus Eradication | cross-sectional study | European Region | United Kingdom | 170 | Clinic | N/A | 56.6% (n = 96) | N/A | N/A | Female, depression. |
| Walter, K. N. et al.[64] (2016) | International Journal of STD & AIDS | cross-sectional study | Region of the Americas | United States | 170 | Hospital | Patients Health Questionnaire version-9 (PHQ 9). | N/A | 35.3% (n = 60) | N/A | Poorer emotional, cognitive quality of life. |
| Alderete, C. et al. (2016) [65] | Salud Mental | cross-sectional study | Region of the Americas | Mexico | 115 | Hospital | 1. The Hospital Anxiety and Depression Scale. 2. Beck Hopelessness Scale. 3. Plutchik Suicide Risk Scale. | 10.4% (n = 12) | N/A | N/A | Depression, Anxiety. |
| Bantjes, J. et al. (2016) [21] | AIDS Care | cross-sectional study | African Region | South Africa | 500 | Clinic | | 24.27% (n = 121) | N/A | N/A | Depression. |
| Bengtson, A. M. et al. (2017) [66] | Journal of Acquired Immune Deficiency Syndromes | cohort study | Region of the Americas | United States | 597 | Database | 1. The Addiction Severity Index (ASI)39 assessed medical, drug, alcohol, employment, legal, family/social, and psychiatric problems. 2. The Functional Assessment of Human Immuno-deficiency Virus Infection quality of life instrument (FAHI). | 38% (n = 227) | N/A | N/A | Depression, mental health, after ART initiation. |
| Carrieri, M. P. et al. (2017) [106] | PLOS ONE | cross-sectional study | European Region | France | 2,973 | Hospital | Patients Health Questionnaire version-9 (PHQ 9). | 6.3% (n = 187) | N/A | N/A | Female, men who have sex with men (MSM), discrimination-related social contexts reported, Homelessness, feeling of loneliness. |
| Croxford, S. et al. (2017) [107] | HIV Medicine | cohort study | European Region | United Kingdom | 88,994 | Database | N/A | N/A | N/A | 1. 2.1 per 10,000 person-years. 2. 1.8% (n = 96) | Male. |
| Egbe, C. O. et al. (2017) [145] | BMC Public Health | cross-sectional study | African Region | Nigeria | 1,187 | Clinic | World Mental Health Composite International Diagnostic Interview (WMH-CIDI) questionnaire | 2.9% (n = 34) | 2.3% (n = 27) | N/A | Alcohol abuse, Depression, marital status, religion. |

(*Continued*)

**Table 1.** (Continued)

| Author, Year of publication | Journal | Type of study | WHO region | Country | Sample size | Study setting | Measurement tool | Suicide ideation rate | Suicide attempt rate | Completed suicide rate | Risk factor |
|---|---|---|---|---|---|---|---|---|---|---|---|
| Ferlatte, O. et al. (2017) [67] | AIDS Care | cross-sectional study | Region of the Americas | Canada | 673 | Database | World Mental Health Composite International Diagnostic Interview (WMH-CIDI) questionnaire. | 22% (n = 150) | 5% (n = 33) | N/A | Rejected as a sexual partner, verbally abused, physically abused. |
| Gebremariam, E. H. et al. (2017) [146] | Psychiatry J | cross-sectional study | African region | Ethiopia | 423 | Hospital | 1. Diagnostic Interview (CIDI) adopted by World. 2. Mental Health (WMH) Survey Initiative version of the World Health Organization (WHO). | 22.5% (n = 95) | 13.9% (n = 59) | N/A | Female, not being on HAART, substance, depression, stigma. |
| Goehringer, F. et al. (2017) [108] | AIDS Research and Human Retroviruses | prospective study | European Region | France | 82,000 | Clinic | N/A | N/A | N/A | 12.5% (n = 10,250) | Socioeconomic difficulty |
| Kalungi, A. et al. (2017) [143] | BMC Genetics | cross-sectional study | African region | Uganda | 600 | Clinic | Mental Health (WMH) Survey Initiative version of the World Health Organization (WHO)." | 3.3% (n = 20) | N/A | N/A | N/A |
| Kinyanda, E. et al. (2017) [148] | Journal of Affective Disorders | cross-sectional study | African region | Uganda | 899 | Clinic | 1. Major Depression disorder (MDD) as defined by DSM IV in the MINI Plus module. 2. Moderate to high risk for suicidality' (MHS) as defined in the suicidality module of the MINI Plus to be a score of nine. | 2.78% (n = 25) | N/A | N/A | Socio-demographic, vulnerability/ protective, stress, impaired psychosocial functioning |
| Lemsalu, L et al. (2017) [109] | AIDS and Behavior | cross-sectional study | African region | Estonia | 828 | Hospital | 1. Composite International Diagnostic Interview (CIDI-10.0). 2. WHO Quality of Life (WHOQOL-HIV-BREF). | 36% (n = 288) | 20% (n = 160) | N/A | Younger age, incarceration, abused alcohol, IDU, having lived with HIV for more than 10 years, depressed. |
| Liu, Y. et al. (2017) [174] | AIDS Care | cross-sectional study | Western Pacific Region | China | 557 | Hospital | N/A | 25% (n = 139) | N/A | N/A | After HIV diagnosis, HIV-related clinical symptoms, stress, Depression, anxiety, social support. |
| Oladeji, B. D. et al. (2017) [149] | Journal of the International Association of Providers of AIDS Care | cross-sectional study | African region | Nigeria | 828 | Clinic | WHO Quality of Life (WHOQOL-HIV-BREF). | 15.1% (n = 125) | 3.9% (n = 32) | N/A | Female, Depression, anxiety. |
| Paparizos, V. et al. (2017) [110] | InfezMed | cross-sectional study | European Region | Greece | 1,884 | Hospital | N/A | N/A | N/A | N/A | Psychiatric co-morbidity, depression, bipolar disorder, anxiety disorder. |
| Quinlivan, E. B. et al. (2017) [68] | AIDS and Behavior | Retrospective study | Region of the Americas | United States | 4,099 | Hospital | Patients Health Questionnaire version-9 (PHQ 9). | 8.6% (n = 352) | N/A | 1.48% (n = 28) | 3 years since HIV diagnosis, HIV RNA >50 copies/ml. |
| Rodriguez, V. J. et al. (2017) [13] | AIDS Care | cross-sectional study | African Region | South Africa | 673 | Clinic | The Edinburgh Postnatal Depression Scale (EPDS-10, α = 0.75). | 39% (n = 262) | N/A | N/A | Female, partner violence, stigma. |

(Continued)

**Table 1.** (Continued)

| Author, Year of publication | Journal | Type of study | WHO region | Country | Sample size | Study setting | Measurement tool | Suicide ideation rate | Suicide attempt rate | Completed suicide rate | Risk factor |
|---|---|---|---|---|---|---|---|---|---|---|---|
| Wang, H. et al. (2017) [175] | Journal of Central South University. Medical sciences | cross-sectional study | Western Pacific Region | China | 504 | Clinic | 1. Self-rating Depression Scale. 2. Beck Scale for Suicide Ideation-Chinese Version. | 27.2% (n = 137) | N/A | N/A | Male, gay, suicide history, anxiety, depression. |
| Wong, M. et al. (2017) [150] | Archives of Women's Mental Health | cohort study | African Region | South Africa | 625 | Clinic | The Edinburgh Postnatal Depression Scale (EPDS) | N/A | 6% (n = 36) | N/A | Age, depression. |
| Woollett, N. et al. (2017) [151] | Journal of Child & Adolescent Mental Health | cross-sectional study | African Region | South Africa | 343 | Hospital | 1. The MINI International Psychiatric Interview for children and adolescents suicide scale. 2. Child Depression Inventory Short Form, which has a high correlation. | N/A | 24% (n = 82) | N/A | Depression, anxiety, PTSD. |
| Ashaba, S. et al. (2018) [152] | Global Mental Health | cross-sectional study | African Region | Uganda | 224 | Hospital | 1. Six-item Internalized AIDS-Related Stigma Scale. 2. Mini International Neuropsychiatric Interview for Children and Adolescents (MINI-KID). | N/A | 13% (n = 29) | N/A | Depression, stigma, bullying. |
| Brennan, C. et al. (2018) [69] | Innovations in Clinical Neuroscience | cross-sectional study | Region of the Americas | Columbia | 1,056 | Clinic | The electronic Columbia-Suicidality Severity Rating Scale (eC-SSRS). | 14% (n = 148) | 9% (n = 96) | N/A | ARTs, depression. |
| Chang, J. L. et al. (2018) [153] | Annals of Internal Medicine | cohort study | African Region | Uganda | 694 | Clinic | N/A | 6.2% (n = 19) | N/A | N/A | Depression. |
| Hentzien, M., A. et al. (2018) [111] | HIV Medicine | cross-sectional study | European Region | France | 349 | Database | N/A | N/A | N/A | 4.1% (n = 99) | Not having children, psychological morbidity, substituted drug consumption, alcohol intake > 20 g/day, abuse, Depression, psychotropic drugs. |
| Hsing-Fei, L. U. et al. (2018) [176] | Journal of Nursing | cross-sectional study | Western Pacific Region | Taiwan | 114 | Hospital | 1. The Beck Scale for Suicidal Ideation (BSS). 2. Beck Depression Index 2nd version (BDI-II). 3. Meaning in Life Questionnaire (MLQ). | 27.2% (n = 31) | 14% (n = 16) | N/A | 1. Suicide ideation: duration since being diagnosed HIV-positive, level of education, depression. 2. Suicide attempt: depression. |
| Loeliger, K. B. et al. (2018) [17] | The Lancet. HIV | case-control study | Region of the Americas | United States | 1,350 | Hospital | N/A | N/A | N/A | 7.6% (n = 13) | Drug overdose, age (>/= 50 years, lower CD4 count (<200 cells per mu), high number of comorbidities, virologic failure, unmonitored viral load. |

(*Continued*)

**Table 1.** (Continued)

| Author, Year of publication | Journal | Type of study | WHO region | Country | Sample size | Study setting | Measurement tool | Suicide ideation rate | Suicide attempt rate | Completed suicide rate | Risk factor |
|---|---|---|---|---|---|---|---|---|---|---|---|
| Loeliger, K. B. et al. (2018) [70] | Topics in Antiviral Medicine | retrospective study | Region of the Americas | United States | 1,350 | Prisons | N/A | N/A | N/A | 8% (n = 13) | Age> = 50 years, lower CD4 count, high number of comorbidities, virologic failure, unmonitored viral load. |
| Lopez, J. D. et al. (2018) [71] | Aids and Behavior | cross-sectional study | Region of the Americas | United States | 648 | Hospital | The Patient Health Questionnaire-9 (PHQ-9). | 13% (n = 81) | N/A | N/A | Anxiety, have unsuppressed viral loads, consider themselves to be homeless. |
| Lu, H. F. et al. (2018) [177] | Hu Li Za Zhi | cross-sectional study | Western Pacific Region | Taiwan | 114 | Prisons | 1. Beck Scale for Suicidal Ideation (BSS,Cronbach's α.89). 2. Beck Depression Index 2nd version (BDI-II, Cronbach's α.89). 3. Meaning in Life Questionnaire (MLQ). 4. The Multidimensional Scale of Perceived Social Support Chinese version, (MSPSS-C, Cronbach's α = .96) | 27.2% (n = 31) | 14% (n = 16) | N/A | Education level, depression. |
| Malava, J. K. et al. (2018) [112] | Malawi Medical Journal | cross-sectional study | European Region | United Kingdom | 206 | Hospital | Patients Health Questionnaire version-9 (PHQ 9). | 16% (n = 33) | N/A | N/A | Alcohol use, depression. |
| Pinto, A. N. et al. (2018) [196] | AIDS Research and Human Retroviruses | cross-sectional study | Western Pacific Region | Australia | 138 | Clinic | N/A | N/A | N/A | 4% (n = 5) | N/A |
| Rodriguez, V. J. et al. (2018) [13] | AIDS and Behavior | cohort study | African region | South Africa | 681 | Database | N/A | 39% (n = 265) | N/A | N/A | Prenatal, intimate partner violence, depression, increased income, stigma, younger age, disclosure of HIV status to partner. |
| Rodriguez, V. J. et al. (2018) [72] | AIDS Care | cross-sectional study | Region of the Americas | Argentina | 118 | Clinic | N/A | 35.6% (n = 42) | N/A | N/A | Female, younger, unemployed, stigma. |
| Sherr, L. et al. (2018) [154] | AIDS Care—Psychological and Socio-Medical Aspects of AIDS/HIV | cross-sectional study | African region | South Africa | 1,058 | Clinic | 1. The Child Depression Inventory short form. 2. Anxiety was measured using the Children's Manifest Anxiety Scale. 3. Posttraumatic stress symptoms. 4. Suicidality/self-harm was measured using the Mini International Neuropsychiatric Interview. | 4.1% (n = 42) | N/A | N/A | Substance use, depression, anxiety. |

*(Continued)*

**Table 1.** (Continued)

| Author, Year of publication | Journal | Type of study | WHO region | Country | Sample size | Study setting | Measurement tool | Suicide ideation rate | Suicide attempt rate | Completed suicide rate | Risk factor |
|---|---|---|---|---|---|---|---|---|---|---|---|
| Sumari-de Boer, M. Et al. (2018) [155] | Tropical Medicine and International Health | cross-sectional study | African region | Tanzania | 365 | Hospital | 1. The Hospital Anxiety and Depression Scale (HADS). 2. The Mini-International Neuropsychiatric Interview (MINI). | 10% (n = 36) | N/A | N/A | Alcohol use, depression. |
| Tang, X. et al. (2018) [12] | Neuropsychiatric Disease and Treatment | cross-sectional study | Western Pacific Region | China | 504 | Clinic | 1. Self-Rating Depression Scale (SDS). 2. Social support rate scale (SSRS, Cronbach's α = 0.762). | 27.2% (n = 137) | N/A | N/A | Depression, social support, emotion-focused coping and problem-focused coping. |
| Wang, W. et al. (2018) [9] | PLOS ONE | cross-sectional study | Western Pacific Region | China | 465 | Hospital | Center for Epidemiological Studies Depression (CES-D, Cronbach's alpha = 0.914). | 31.6% (n = 147) | N/A | N/A | Low social support, depression, stigma, older age, low education level, married, having children, psychosocial. |
| Zarei, N. et al. (2018) [186] | AIDS Research and Treatment | cross-sectional study | Eastern Mediterranean region | Iran | 351 | Hospital | To evaluate stigma and discrimination, 12 items (four-choice question, 7 for internal and 5 for external/ social stigma). | 15.4% (n = 54) | N/A | N/A | Quality of life, spiritual beliefs, stigma, age, gender, marital status. |
| Zeng, C. et al. (2018) [178] | BMC Public Health | cross-sectional study | Western Pacific Region | China | 411 | Clinic | The Center for Epidemiologic Studies Depression Scale (CES-D), a 20-item scale with four dimensions, including depressed affect, positive affect, somatic and retarded activity, and interpersonal problems. Internal consistency estimate of reliability of the scale was good (Cronbach's alpha = 0.93). | 32.4% (n = 133) | 9% (n = 37) | N/A | Depression. |
| Casale, M. et al. (2019) [156] | Journal of Affective Disorders | cross-sectional study | African region | South Africa | 1,053 | Hospital | 1. Mini International Psychiatric Interview for Children and Adolescents Suicidality and Self-harm subscale. 2. Children's Depression Inventory (CDI 10-item short form) α = 0.64. | 6.2% (n = 66) | N/A | N/A | Depression, stigma. |
| Croxford, S. et al. (2019) [113] | HIV Medicine | retrospective study | European Region | United Kingdom | 166 | Database | N/A | N/A | N/A | 7% (n = 12) | N/A |
| Knieps, L. et al. (2019) [114] | HIV Medicine | cohort study | European Region | Germany | 81 | Database | N/A | N/A | N/A | 6.2% (n = 5) | N/A |
| Kreniske, P. et al. (2019) [17] | Journal of Adolescent Health | cross-sectional study | Region of the Americas | United States | 340 | Clinic | N/A | N/A | NA | N/A | Age 14–18. |
| Li, Y. et al. (2019) [179] | Journal of Medical Internet Research | longitudinal study | Western Pacific Region | Taiwan | 300 | Hospital | 1. The Chinese version of the CES-D scale. 2. Perceived Stress Scale (PSS-10). | 45% (n = 135) | 9.7% (n = 29) | N/A | Stress, depression. |

*(Continued)*

**Table 1.** (Continued)

| Author, Year of publication | Journal | Type of study | WHO region | Country | Sample size | Study setting | Measurement tool | Suicide ideation rate | Suicide attempt rate | Completed suicide rate | Risk factor |
|---|---|---|---|---|---|---|---|---|---|---|---|
| Lu, H. F. et al. (2019) [180] | Journal of advanced nursing | longitudinal study | Western Pacific Region | Taiwan | 113 | Hospital | 1. Beck Depression, Inventory-II (BDI-II). 2. Meaning in Life Questionnaire and the Multidimensional Scale of Perceived Social Support at baseline. | 27.2% (n = 31) | 14.7% (n = 17) | N/A | Education level, social support from family, depression. |
| Mandell, L. N. Et al. (2019) [73] | AIDS and behavior | cross-sectional study | Region of the Americas | Argentina | 360 | Hospital | N/A | 21% (n = 76) | N/A | N/A | Younger age, depression. |
| Peltekis, A. et al. (2019) [115] | Psychiatrike = Psychiatriki | cross-sectional study | European Region | Greece | 191 | Clinic | Patients Health Questionnaire version-9 (PHQ 9). | 9.2% (n = 17) | 14% (n = 27) | N/A | N/A |
| Ruffieux, Y. et al. (2019) [116] | Journal of the International AIDS Society | longitudinal study | European Region | Switzerland | 20,136 | Hospital | N/A | N/A | N/A | 1% (n = 204) | Gender, male, nationality, centers for disease control and prevention clinical stage, IDU, mental health, psychiatric treatment. |
| Sarna, A. et al. (2019) [181] | Archives of Women's Mental Health | retrospective study | South-East Asian region | India | 200 | Clinic | N/A | N/A | 23% (n = 46) | N/A | Depression. |
| Shim, E. J. et al. (2019) [181] | International journal of behavioral medicine | cohort study | Western Pacific Region | South Korea | 202 | Database | 1. The Hospital Anxiety and Depression Scale. 2. Mini-International Neuropsychiatric Interview suicidality module. | 20% (n = 40) | N/A | N/A | Thwarted belongingness (TB), perceived burdensomeness (PB), depression. |
| Tyree, G. A. et al. (2019) [74] | Journal of Affective Disorders | cross-sectional study | Region of the Americas | United States | 1,002 | Clinic | Patient Health Questionnaire 9 (PHQ-9) | 38.4% (n = 385) | N/A | N/A | Depression. |
| Wang, Y. Y. et al. (2019) [182] | Psychology, health & medicine | cross-sectional study | Western Pacific Region | China | 410 | Clinic | Generalized Anxiety Disorder score (GAD-7) | 10.7% (n = 44) | 3.2% (n = 13) | N/A | Unemployment, age, CD4 lymphocyte counts, anxiety. |
| Wonde, M. et al. (2019) [75] | PLoS ONE | cross-sectional study | Region of the Americas | United States | 413 | Hospital | Patient Health Questionnaire 9 (PHQ-9) | 27.1% (n = 112) | 16.9% (n = 70) | N/A | 1. Suicide ideation: Female, family death, WHO clinical stage III of HIV, WHO clinical stage IV of HIV, depression, perceived HIV stigma. 2. Suicide attempt: female, opportunistic infections, WHO clinical stage III of HIV, depression, poor social support. |

*(Continued)*

Table 1. (Continued)

| Author, Year of publication | Journal | Type of study | WHO region | Country | Sample size | Study setting | Measurement tool | Suicide ideation rate | Suicide attempt rate | Completed suicide rate | Risk factor |
|---|---|---|---|---|---|---|---|---|---|---|---|
| Adeyemo, S. et al. (2020) [157] | Child and Adolescent Psychiatry and Mental Health | cross-sectional study | African region | Nigeria | 201 | Hospital | Mini International Neuropsychiatric Interview for children and adolescents (MINI-Kid). | 35.3% (n = 71) | N/A | N/A | High ACE score, physical abuse, emotional abuse, depression. |
| Bi, F. Y. et al. (2020) [183] | Psychology Health & Medicine | cross-sectional study | Western Pacific Region | China | 557 | Clinic | 1. Social Support Rating Scale (SSRS). 2. Chinese HIV/AIDS Stress Scale (CSS-HIV). 3. Patient Health Questionnaire (PHQ-9). | 24.95% (n = 139) | N/A | N/A | Low social support, depression. |
| Durham, M. D. et al. (2020) [76] | Preventive Medicine | prospective study | Region of the Americas | United States | 6,706 | Hospital | N/A | 3.3% (n = 224) | N/A | N/A | <50 years old, non-Hispanic/Latino black, have CD4+ cell count <350 cells/mm3, have a viral load ≥50 copies/mL, have stopped antiretroviral therapy, alcohol dependence, drug overdose. |
| Fontela, C. et al. (2020) [117] | Scientific reports | Cohort study | European Region | Spain | 1,059 | Database | N/A | N/A | N/A | 1.32% (n = 14) | N/A |
| Kindaya, G. G. et al. (2020) [13] | HIV/AIDS—Research and Palliative Care | cross-sectional study | African region | Ethiopia | 412 | Hospital | N/A | 24.3% (n = 100) | 12.6% (n = 52) | N/A | 1. Suicide ideation: extreme poverty, living alone, widowed, CD4 level less than 250, current alcohol use. 2. Suicidal attempt: urban residence, stage IV HIV, family history of suicide, depression. |
| Knettel, Brandon A. (2020) [158] | AIDS | cross-sectional study | African region | Tanzania | 200 | Clinic | N/A | 12.8% (n = 26) | N/A | N/A | Anxiety, HIV stigma. |
| Mebrahtu, H. et al. (2020) [159] | AIDS & Behavior | cross-sectional study | African region | Zimbabwe | 562 | Hospital | 1. Parental stress index-short form (PSI-SF). Common mental disorders (CMDs). | 30.43% (n = 171) | N/A | N/A | Younger, unmarried, experienced moderate to severe hunger, had elevated parental stress, depression symptoms. |
| Nishijima, T. et al. (2020) [184] | AIDS | cohort study | Western Pacific Region | Japan | 277 | Clinic | N/A | N/A | N/A | 7.76% (n = 14) | N/A |
| Ophinni, Y. et al. (2020) [10] | BMC Psychiatry | cross-sectional study | South-East Asian region | Indonesia | 86 | Clinic | Symptom Checklist-90 (SCL-90). | 23.3% (n = 20) | N/A | N/A | Depression, anxiety, non-marital status, CD4 count < 500 cells/µl, efavirenz use. |

(*Continued*)

**Table 1.** (Continued)

| Author, Year of publication | Journal | Type of study | WHO region | Country | Sample size | Study setting | Measurement tool | Suicide ideation rate | Suicide attempt rate | Completed suicide rate | Risk factor |
|---|---|---|---|---|---|---|---|---|---|---|---|
| Sereda, Y. et al. (2020) [118] | Journal of the International AIDS Society | cross-sectional study | European Region | Ukraine | 191 | Hospital | 1. Primary outcomes were HIV stigma (Berger scale). 2. Substance use stigma (Substance Abuse Stigma Scale). | N/A | 28% (n = 54) | N/A | HIV stigma. |
| Gizachew et al. (2021) [196] | Annals of General Psychiatry | cross-sectional study | African region | Ethiopia | 326 | Hospital | 1. Composite International Diagnostic Interview (CIDI). 2. Patient Health Questionnaire for Anxiety and Depression (PHQ-4). | 16% (n = 52) | 7.1% (n = 23) | N/A | Low monthly income, living alone, suicidal thought before knowing seropositive status, family history of suicide, experiencing mild and moderate-to-severe depression and anxiety symptoms, use of khat. |
| Tamirat et al. (2021) [197] | HIV/AIDS | cross-sectional study | African region | Ethiopia | 395 | Hospital | 1. Composite International Diagnostic Interview (CIDI). 2. Patient Health Questionnaire (PHQ-9). | 9.40% (n = 37) | 3.3% (n = 13) | N/A | Low body mass index, stages three and above illnesses, depression, poor social support, and fair and poor adherence. |
| Wang et al. (2021) [199] | Journal of Affective Disorders | cross-sectional study | Western Pacific Region | China | 494 | Hospital | Personal Social Capital Scale (PSCS-8). | 32.59% (n = 161) | 12.15% (n = 60) | N/A | Social capital. |
| Zewdu et al. (2021) [198] | BMC Pregnancy and Childbirth | cross-sectional study | African region | Ethiopia | 414 | HIV clinics | Composite International Diagnostic Interview (CIDI). | 8.20% (n = 34) | N/A | N/A | Perinatal depression, not disclosed HIV status, unplanned pregnancy. |

### Qualitative synthesis

Qualitative synthesis was conducted using data extraction findings to explore the key themes within the selected studies. Three researchers independently conducted the qualitative synthesis on the baseline risk factors of the participants and suicidal ideation rate, suicide attempt rate, and completed suicide rate (Table 1). We created a structural model related to the consistent risk factors after final agreement among all authors.

## Results

### Study identification

After searching the six databases, 8,055 articles published from January 1, 1988, to July 8, 2021, were identified (Ovid Medline, 878; Embase, 2,123; CINAHL, 815; Web of Science, 2,450; Academic Search Complete, 1,357; Psychology & Behavioral Sciences Collection, 388; manual search, 44). After removing 3,205 duplicate articles, 4,850 articles were screened for the title and abstract. Of these, 1,954 met the inclusion criteria and they were eligible to be considered for reading in the systemic review. The remaining 2,896 articles were excluded for the following reasons: 1,716 did not mention HIV/AIDS; 820 did not mention suicidal behavior; 262 did not clearly assess the outcome variables; and 98 were not available in full text format. After quality assessment, 193 articles were included in this scoping review (Fig 1). All included studies were published as a full article in peer-reviewed journal.

### Characteristics of the included studies

According to the WHO regions [22], 69 studies were performed Region of the Americas [9, 17, 20, 24–80], followed by 45 in Europe Region [5, 29, 31, 77–121], 45 in Africa Region [13, 18, 21, 126–164], 26 in Western Pacific Region [9, 165–190], 2 in the Eastern Mediterranean Region [191, 192], and 6 South-East Asian Region[9, 16, 168, 169, 171, 181] (Fig 2A). A total of 130 articles were published during the past 10 years (2011–2021) (Fig 2B).

There were 121 studies conducted with cross-sectional study designs, 37 cohort studies, 24 retrospective studies, six longitudinal studies, three case-control studies, and two prospective studies (Fig 2C). According to the study settings, 88 studies were conducted in a hospital setting, 35 referenced databases, 64 were conducted in clinics or community centers, 4 were performed in prisons, and 2 were done at palliative care centers (Fig 2D).

107 articles used measurement tools to identify the condition of suicidal behavior and risk factors. There were 66 articles on suicidal ideation, 38 on suicidal attempts, 47 on completed suicide, 39 on suicidal ideation and attempts, 1 on suicidal ideation and death by suicide [95], and three on suicidal ideation, suicide attempts, and completed suicides [78, 79, 91]. Most of the studies evaluated risk factors; however, 23 did not mention risk factors (Table 1).

### Epidemiology of suicidal behaviors among PLHIV

**Suicidal ideation around the world.** The Pre HAART era from 1990 to 1996, suicidal ideation prevalence rate is 28.6 to 55% in United States [24, 27, 187, 188]; 50.5% in United Kingdom [79]; and 1.18% in Spain [78].

In the Post-HAART era, from 1997 to 2021, the suicidal ideation prevalence rate was 60% to3.3% in the United States [10, 20, 34–36, 39, 41, 43, 44, 45–49, 55, 63, 66, 68, 71, 74–76, 133]; 69% to 16% in the United Kingdom [80, 105, 112]; 13.2% to 6.3% in France [82, 85, 95, 106]; 9.2% in Greece [115]; 11.4% to 6.2% in Africa [13, 21, 125, 126, 130, 134, 135, 138, 154, 156]; 8.8% to 6.2% in Uganda [142–144, 147, 148, 153]; 34.7% to 35.3% in Nigeria [129, 145, 149, 157]; 35.6% to 21% in Argentina [75, 77]; 67% in Australia[195]; 50% in Brazil [16, 17]; 9.2%

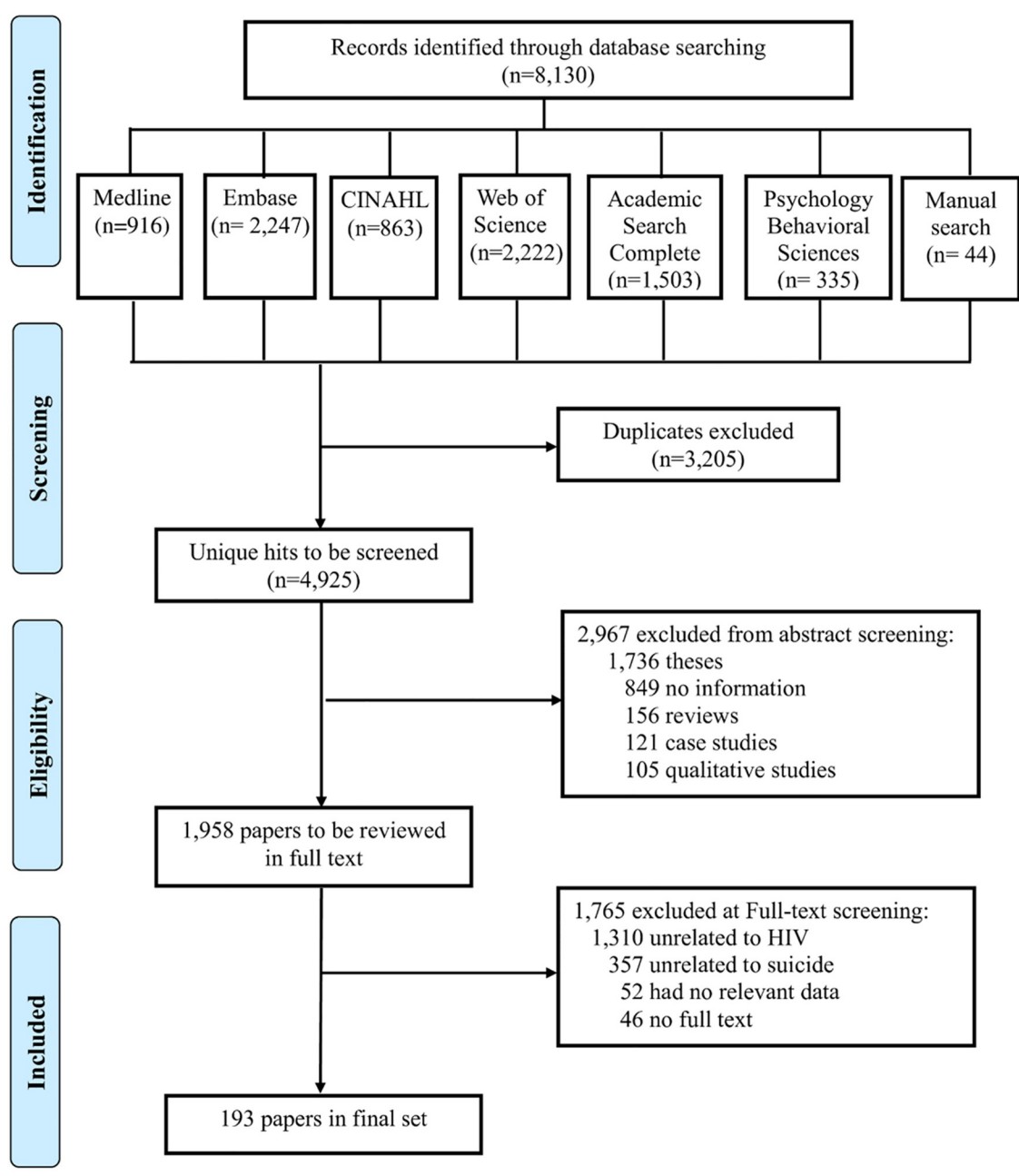

**Fig 1. PRISMA style flow diagram included studies.**

to 22% in Canada [50, 67]; 14% in Columbia [69]; 36% in Estonia [109]; 33.6% to 8.2% in Ethiopia [13, 139, 146, 190–192]; 14% in Nepal [174, 177]; 48% in the Netherlands [99]; 34% to 32.6% in China [9, 12, 164, 167, 174, 175, 178, 182, 183, 193]; 39.3% to 27.2% in Taiwan [13, 161, 176, 177, 179, 180]; 10% to 12.8% in Tanzania [155, 158]; to 23.3% in Indonesia [9]; 10.4% in Mexico [65]; 7.1% in Malawi [137]; 12.1% in Romania [92]; 56% in Russia [98]; 44% to 20% in Korea [173, 181]; 15.4% in Iran [186]; and 30.4% in Zimbabwe [159] (S1 Table).

**Suicide attempts around the globe.** In the Pre-HAART era, from 1994 to 1996, the suicidal attempt prevalence rate was 21.8% to 42.8% in the United States [28, 160]; 21.4% in the United Kingdom [79]; and 4.02% in Spain [78].

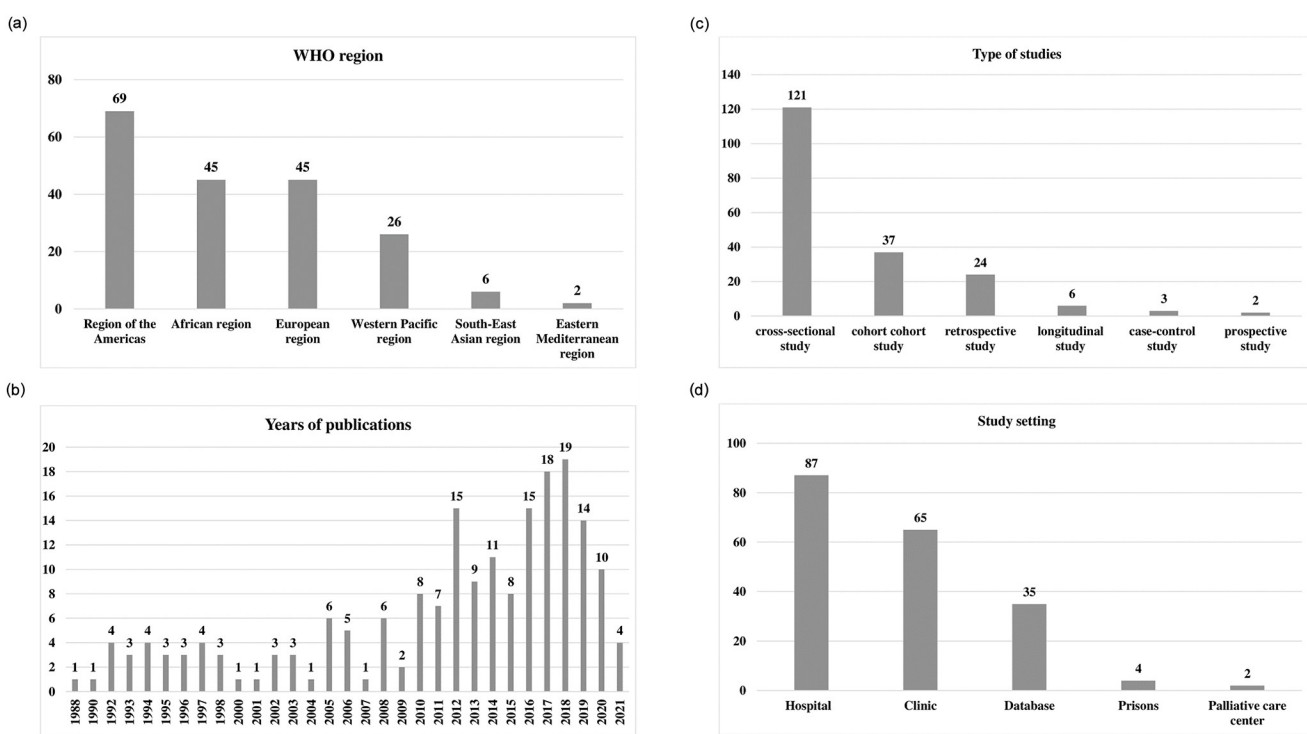

**Fig 2.** A. Studies distribution according to world health organization (WHO) regions. B. Years of publications. C. Types of studies. D. Study setting.

In the Post-HAART era, from 1997 to 2021, the suicidal attempt prevalence rate was 21% to 16.9% in the United States [20, 37, 38, 43–45, 45, 53, 54, 56, 59, 64, 75]; 31% to 6.6% in the United Kingdom [80, 89, 91, 93, 96, 97]; 22% in France [120, 121]; 14% in Greece [115]; 72% in Spain [103]; 54% to 6% in Africa [123, 127, 135, 150, 151]; 17.4% to 13% in Uganda [124, 128, 132, 142–144, 152]; 9.3% to 2.3% in Nigeria [18, 129, 145, 149]; 27% to 85.2% in Australia [189, 194]; 18% in Brazil [62]; 10.3% to 5% in Canada [50, 67]; 9% in Columbia [69]; 20% in Estonia [109]; 17% in Nepal [169]; 34% in the Netherlands [99]; 17% in the Nepal [169]; 8% to 12.2% in China [164, 178, 179, 182, 193]; 26.7% to 9.7% in Taiwan [162, 165, 166, 176, 177, 179, 180]; 23% in India [181]; 36% in Russia [98]; 11% in Korea, 28% in Ukraine [118]; and 20.4% in Puerto Rico [58] (S2 Table).

**Distribution of death due to suicide around the world.** In the Pre-HAART era, from 1988 to 1996, the completed suicide incidence rate was 680.56–4.9per 100,000 person years in the United States [23, 25, 30, 32, 195]; 0.5% in the United Kingdom [79]; 25% in the Sweden [77]; 0.47% in the Spain [78]; and 8.13% to 13% in the Netherlands [29, 31]. The death rate due to suicide went from 0.68 per 100 person-years in 1988 to 0.05 in 1996 in the United States [23, 25, 30, 32].

In the Post-HAART era, from 1997 to 2020, the completed suicide prevalence rate was 8.7% to 7.6% in the United States [17, 33, 47, 52, 70, 84, 86, 87]; 12.9% to 7% in the United Kingdom [88, 91, 102, 104, 107, 113]; 6% to 4.1% in France [81, 83, 95, 108, 111]; 1.48% in Greece [110]; 0.6%–1.3% in Spain [90, 117]; 4% to 6.2% in Germany [114, 136]; 6% to 1% [104, 105, 120]; 4% in Australia [196]; 7% to 8.2% in Canada [16, 61]; 7.8% in Japan [184]; 5.5% in Taiwan [163]; 15.5% in Thailand [168]; and 38.6% in Denmark [94]. The death due to suicide incidence rate went from 0.03 per 100 person years in 2010 to 0.47 in 2016 in Canada [16, 61], 0.16 per 100 person-years in Switzerland [5], and was 0.04 per 100 person years in 2015 to 0.02 in 2017 in UK [102, 107] (S3 Table).

### Risk factors for suicide behavior among PLHIV

In this review, the identified causes of death by suicide included drug overdose, gunshot, jumping, drug poisoning, suffocation, and cutting wrists (Table 2). Additionally, we found that suicide-related risk factors included demographic, physiological, social, environmental, and psychological factors (Fig 3).

Among the demographic factors, there were inconsistent risk factors for suicidal behavior such as gender (male [5, 33, 34, 81, 90, 111, 120], female [41, 92, 105]), age (young age [13, 16, 75, 77, 113, 138, 164], middle age[10, 18, 80, 142, 178], older age [9]), sexual orientation, education level, In respect to specific psychological symptoms and disorders, depression, substance abuse, anxiety, intravenous drug use, post-traumatic stress disorder, major mood disorders, and mental disorders were found to be consistent suicide risk factors [21, 22, 56, 65, 68, 74, 78, 97, 102, 116, 132, 135, 138, 141, 145–147, 149, 154, 155, 157, 158, 159, 172, 173, 154, 187, 192]. Among the physiological factors were HAART side effects, poor immune status, physical symptoms, comorbid illnesses, insomnia, CD4 cell count, unmonitored viral load, neurocognitive developmental disorders, opportunistic infections, and medical status; an inconsistent risk factor was HIV exposure time. The social factors of quality of life, living alone, less coping self-efficacy, violence, bullying, incarceration, and bereavement were inconsistent risk factors for suicidal behavior. However, low social support was a consistent risk factor. Among the environmental factors, socioeconomic status, ethnicity, and having children were inconsistent risk factors; however, discrimination and religion [142, 149, 193] were consistent risk factors (Table 2 and Fig 3).

### Measurement tools of the suicide behaviors suicidal behavior and risk factors

Within the included 193 studies, we found that 12 different scales were used to measured suicidal behavior and its risk factors; 26 studies used the Beck depression inventory scale, 8 used the Beck scale for suicide ideation, and 4 used the five-item brief symptom rating scale (Table 3).

### Discussion

About 40 million people of the global population are currently living with HIV/AIDS. The era of HAART treatment has brought significant improvements in patient longevity and quality of life [202]; however, PLHIV experience a heavy burden of psychosocial conditions that are frequently undiagnosed and untreated. The pooled incidence of suicide completion among PLHIV globally was 10.2 per 1000 population, translating to a 100-fold greater suicide completion rate compared with the global population rate of 0.09/1000 population for 2019 [3, 203]. Therefore, this scoping review of 193 studies included an overview of three types of suicidal behavior among PLHIV as follows: suicidal ideation, suicidal attempts, and dying by suicides. We also included risk factors and associations of suicidal behavior according to demographic, social, physiological, psychological, and environmental factors. We identified consistent and inconsistent risk factors among the three types of suicidal behavior (Fig 3).

In total, this review encompasses 729,189 participants from 49 countries with all eligible articles published during the past 33 years (1988 to 2021). Two-thirds of the studies were published in the last five years (80/193). We found that there was an increasing trend toward conducting research related to suicidal behavior and risk factors among PLHIV globally.

Most studies were conducted in the United States or were performed by researchers from the United States conducting research in other countries, especially in Africa or developing countries (i.e., Nepal and Thailand). This is likely due to global funding strategies and

**Table 2. Risk factors of suicide among people living with HIV.**

| Risk factors | Suicide type | Suicidal ideation | Suicide attempt | Suicidal complete |
|---|---|---|---|---|
| Demographic | Gender (♂&♀) inconsistent | Gender: 1 article [192] | Gender:1 article [61] | Gender: 1 articles [64] |
| | | Male:6 articles [9, 40, 49, 131, 176, 180] | Male: 3 articles [38, 100, 125] | Male: 7 articles [5, 33, 34, 81, 90, 111, 120] |
| | | Female:15 articles [14, 16, 18, 45, 75, 79, 84, 109, 110, 133, 137, 143, 146, 150, 153] | Female: 13 articles [15, 43, 45, 79, 84, 124, 132, 133, 136, 143, 146, 150, 153] | Female: 3 articles [41, 92, 105] |
| | Age (Young, middle, and old age) inconsistent | Age (all ages):2 article [188, 192] | Age (all ages): 1 article [188] | Age (all ages): 2 articles [26, 64] |
| | | Young: 7 articles [13, 16, 75, 77, 113, 138, 164] | Young: 2 articles [58, 113] | Young: 1 article [60] |
| | | Middle: 5 articles [10, 18, 80, 142, 178] | Middle: 3 articles [43, 124, 178] | Middle: 4 articles [5, 24, 34, 73] |
| | | Older:1 article [9] | Older: 0 article | Older: 2 articles [17, 105] |
| | Sexual orientation inconsistent | Sexual orientation: 2 articles [37, 70] | Sexual orientation: 3 articles [58, 61, 70] | Sexual orientation: 1 article [81] |
| | | Homosexual: 4 articles [40, 110, 180, 195] | Homosexual: 1 article [195] | Homosexual: 0 article |
| | | Heterosexual:1 article [49] | Heterosexual: 0 article | Heterosexual: 0 article |
| | Low education level | 5 articles [9, 142, 181, 181, 185] | 4 articles [124, 181, 181, 185] | 0 articles |
| | Marital status inconsistent | Single: 2 articles [11, 164] | Single: 0 article | Single: 1 article [175] |
| | | Married: 4 articles [57, 143, 149, 192] | Married: 3 articles [107, 143, 149] | Married: 1 article [63] |
| | | Widow: 1 article [162] | Widow: 1 article [162] | Widow: 0 article |
| Psychological symptoms & disorders | Stigma | 14 articles [9, 12, 13, 14, 42, 75, 102, 143, 148, 150, 160, 163, 188, 192] | 6 articles [79, 102, 143, 148, 150, 156] | 0 article |
| | Psychological symptoms consistent | 9 articles [9, 37, 49, 115, 144, 171, 178, 179, 194] | 3 article [56, 171, 178] | 1 article [115] |
| | Stress | 7 articles [42, 57, 152, 164, 169, 179, 184] | 5 articles [125, 132, 169, 175, 184] | 0 article |
| | Hopelessness | 3 articles [110, 148, 194] | 2 articles [58, 148] | 0 article |
| | Anger | 2 articles [103, 148] | 2 articles [103, 148] | 0 article |
| | Perceived burdensomeness (PB) | 2 articles [35, 132, 187] | 0 article | 0 article |
| | Fear | 1 article [103] | 1 article [103] | 0 article |
| | Guilt | 1 article [103] | 1 article [103] | 0 article |
| | Depression consistent | 59 articles [9–11, 18, 21, 22, 25, 28, 39, 42, 45, 48, 50, 57, 66, 68, 69, 72, 75, 77–79, 89, 99, 102, 103, 109, 113, 116, 119, 129, 130, 134, 135, 137, 139, 141, 143, 146, 147, 149, 150, 152, 153, 157–164, 169, 170, 172, 174, 176–185, 187, 189, 193, 194] | 36 articles [21, 43, 45, 65, 72, 79, 82, 93, 102, 103, 113, 125, 128, 136, 139, 143, 145, 146, 148–150, 153, 156, 162, 169–171, 175, 178, 181, 181, 183–186, 200] | 5 articles [41, 105, 114, 115, 173] |
| | Substance abuse consistent | Substance abuse: 10 articles [9, 18, 21, 28, 52, 96, 102, 150, 158, 159] | Substance abuse: 9 articles [21, 39, 52, 55, 56, 102, 107, 125, 150] | Substance abuse: 1 article [115] |
| | | Drug abuse: 1 article [48] | Drug abuse: 2 articles [101, 102] | Drug abuse: 0 article |
| | | Alcohol abuse: 5 articles [80, 113, 116, 149, 162] | Alcohol abuse: 5 articles [101, 102, 113, 132, 149, 154, 162, 175] | Alcohol abuse: 1 article [88] |
| | Anxiety consistent | 16 articles [11, 12, 18, 28, 45, 68, 74, 148, 153, 158, 163, 169, 170, 179, 180, 188] | 9 articles [45, 93, 119, 125, 139, 148, 153, 155, 169–171, 188] | 1 article [114] |
| | Intravenous drug-using (IDU) consistent | 3 articles [82, 113, 195] | 6 articles [55, 82, 113, 125, 175, 195] | 11 articles [5, 30, 41, 59, 60, 63, 64, 82, 94, 105, 120] |
| | Post-traumatic stress disorder (PTSD) | 1 article [146] | 2 articles [127, 146] | 0 article |
| | Psychiatric consistent | 4 articles [36, 57, 69, 96] | 4 articles [38, 132, 136, 170] | 3 articles [98, 114, 120] |
| | Major mood disorder | 3 articles [21, 28, 182] | 2 articles [19, 21] | 1 article [120] |

*(Continued)*

**Table 2.** (Continued)

| Risk factors | Suicide type | Suicidal ideation | Suicide attempt | Suicidal complete |
|---|---|---|---|---|
| Physiological | HIV exposure time consistent | HIV exposure time: 1 article [49] | HIV exposure time: 0 article | HIV exposure time: 1 article [201] |
| | | Early time: 4 articles [71, 83, 147, 179] | Early time: 3 articles [83, 125, 147] | Early time: 4 articles [33, 83, 88, 98] |
| | | Long time: 4 articles [16, 79, 113, 162] | Long time: 3 articles [79, 113, 162] | Long time: 1 article [5, 111] |
| | HAART side effect consistent | ART (efavirenz): 4 articles [11, 69, 72, 174] | ART (efavirenz): 3 articles [62, 72, 174] | ART (efavirenz): 0 article |
| | | HAART side effect: 3 articles [49, 80, 151] | HAART side effect: 0 article | HAART side effect: 1 article [166] |
| | | Not side effect: 1 article [150] | Not side effect: 1 article [150] | Not side effect: 0 article |
| | Physical symptoms consistent | 4 articles [37, 49, 147, 152] | 2 articles [147, 171] | 1 article [73] |
| | CD4 cell count consistent | 5 articles [11, 80, 96, 143, 162, 188, 195] | 3 articles [162, 165, 188] | 5 articles [17, 60, 64, 73, 90] |
| | Unmonitored viral load | 2 articles [74, 80] | 0 article | 2 articles [17, 73] |
| | Opportunistic infection | 3 articles [79, 143, 178] | 3 articles [79, 143, 178] | 0 article |
| | Medical status | 0 article | 1 article [19] | 1 article [98] |
| Social support | Low social support consistent | 13 articles [9, 35, 37, 79, 143, 148, 177, 178, 179, 182, 185, 189, 194] | 8 articles [58, 67, 79, 143, 148, 175, 178, 185] | 1 article [81] |
| | Quality of life | 2 articles [49, 192] | 2 articles [19, 67] | 0 article |
| | Living alone | Living alone: 4 articles [110, 162, 178, 195] | Living alone: 3 articles [162, 178, 195] | 0 article |
| | | Not living alone: 0 article | Not living alone: 0 article | 0 article |
| | Less coping self-efficacy | 1 article [42] | 1 article [132] | 0 article |
| | Violent | 6 articles [13, 14, 16, 28, 70, 161] | 1 article [70] | 0 article |
| | Bullying | 0 article | 1 article [156] | 0 article |
| | Incarceration | 1 article [113] | 1 article [113] | 0 article |
| | Bereavement | 2 articles [35, 79] | 1 article [79] | 0 article |
| Environment | Economic status inconsistent | Employment: 0 article | Employment: 2 articles [15, 61] | Employment: 1 article [105] |
| | | Unemployment: 14 articles [9, 13, 18, 49, 50, 58, 75, 76, 142, 144, 147, 148, 164, 195] | Unemployment: 9 articles [19, 38, 124, 132, 147, 148, 162, 188, 195] | Unemployment: 3 articles [92, 98, 112] |
| | Race inconsistent | White: 4 articles [9, 40, 47, 49] | White: 1 article [47] | 1 article [26] |
| | | Black: 2 articles [49, 80] | Black: 0 article | Black: 0 article |
| | Religion | 3 articles [142, 149, 192] | 2 articles [19, 149] | 0 article |
| | Having children inconsistent | Having children: 1 article [9] | Having children: 1 article [15] | Having children: 0 article |
| | | Not having children: 0 article | Not having children: 0 article | Not having children: 1 article [115] |
| | Discrimination | 1 article [110, 176] | 0 article | 0 article |
| Cause of death | Drug overdosage consistent | 4 articles [28, 80, 82, 83] | 2 articles [82, 83] | 7 articles [17, 26, 30, 54, 81–83] |
| | Drug poisoning | 0 articles | 1 article [26] | 1 article [26] |
| | Firearms consistent | 1 article [28] | 1 article [26] | 1 article [26] |
| | Jumping | 1 article [28] | 1 article [26] | 0 article |
| | Cutting wrists | 1 article [28] | 1 article [26] | 0 article |
| | Suffocation. | 0 article | 1 article [26] | 1 article [26] |

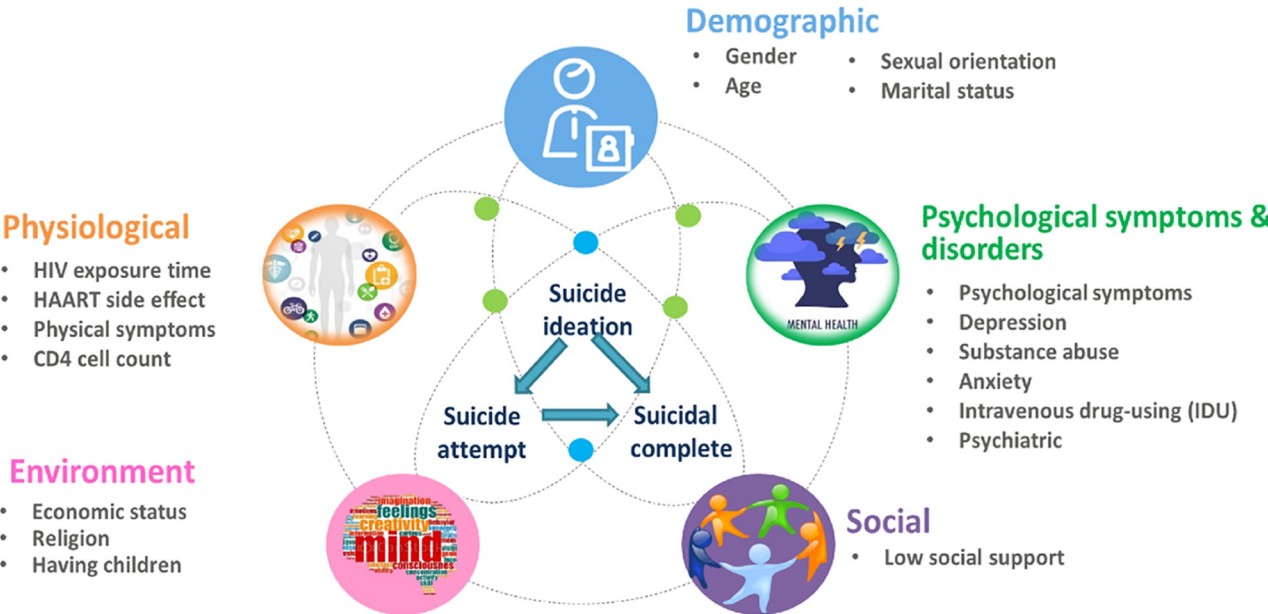

**Fig 3. Relationship of suicide type and risk factor among people living HIV as a model there are 5 variations in people living with hiv suicide model.** The factors are from risk factors in data which study suicide ideation, suicide attempt and completed suicide.

continuing education programs conducted by United States universities along with partnership programs in other countries[14, 44, 75, 127, 129, 137, 138, 140–142, 144, 145, 154, 156, 157, 163, 186]. Most studies were conducted in hospitals and clinics. However, long-term observational data were extracted from databases as well, and of four studies conducted in prisons, three were in Taiwan [170, 171, 181], and one were in United States [73].

According to the findings of this review, the prevalence rate was highest in the United States, United Kingdom, Australia, and Russia for suicidal ideation, suicide attempts rate was

**Table 3. Measurement instrument of suicide.**

| Categories | Measurement tool | Studies |
|---|---|---|
| 12 suicide scales | Beck depression inventory (BDI) | 26 |
| | [12, 18, 20, 24, 37, 46, 62, 85, 98, 126, 129–134, 137, 138, 164, 167, 169, 172, 176, 177, 180, 187] | |
| | Beck scale for suicide ideation (BSS) [16, 36, 46, 99, 175–177, 150] | 8 |
| | The five-item brief symptom rating scale (BSRS-5) [18, 159, 165, 166] | 4 |
| | Plutchik suicide risk scale [96] | 1 |
| | Positive and negative suicide ideation (PANSI) [51] | 1 |
| | GAIN scales: Suicidal/Homicidal thought scale [49] | 1 |
| | Harkavy Asnis Suicide Survey scale. [15] | 1 |
| | Suicide ideation self-report scale [36] | 1 |
| | Suicide ideation questionnaire (QIS) [16] | 1 |
| | Mini international psychiatric interview for children and adolescent's suicidality and self-harm subscale [15] | 1 |
| | Suicide assessment questions from the national institute of mental health diagnostic interview schedule version III-A (DIS) [43] | 1 |
| | The electronic Columbia-suicidality severity rating scale (eC-SSRS) [69] | 1 |

highest in the United States, Australia, and Spain, and death due to suicide rate was highest in Denmark, and Thailand among PLHIV from 2000 to 2020. The highest suicide ideation rate was in the UK [84], followed by Australia [195], and the US [37]. The highest suicide attempt rate was in Australia [195], with the second highest in Spain [107], and third in South Africa [127]. The highest completed suicide rate was in Denmark [98], followed by Thailand [173], and France [112]. These findings may be since these countries have the most liberal laws on doctor-assisted suicide or gun control or could be due to economic recessions and societal pressure [26, 28, 204, 205]. These differences could be attributed to discrepancies in cultural differences, religious dimensions, and socioeconomic status, and not just by geographical location alone [4]. Previous research has identified psychological disorders and suicide are extremely connected and established in high-income countries, with many suicides occurring impulsively in moments of crisis with a breakdown in the ability to deal with life stresses. This review also found similar results[206, 207].

The most frequently used methods used of suicide are hanging and pesticide poisoning in Western countries[17, 26, 28, 30, 54, 80–83]. Reported risk factors for suicide attempts include mental and physical health problems, socioeconomic problems, and drug and alcohol use/ abuse [208] According to our finding in when we considered about South-East Asian Region, most common suicide behavior is death due to suicide, compared with suicide attempts and suicide ideation. Because of educational status of family and social pressure also the social discrimination and stigma are more common in Asian countries than elsewhere in the world [206, 207].

Depression and suicidal thinking occur frequently alongside HIV/AIDS, triggering profound detrimental impacts on quality of life, treatment adherence, disease progression, and mortality [177, 209]. According to this scoping review, 85 articles dealing with depression, the most common death-related factor for PLHIV is suicide ideation, and their attempted suicide behavior risk is due to depression which is its common cause. Bullying, which includes stigmatization and discrimination, can also drive people to suicide as it increases social isolation. Substance abuse and overdose or severe physical disease are also recognized causes. According to this review findings, Caribbean countries and the Middle East showed the lowest death rates due to suicide.

This study is a global overview of suicidal behavior and associated risk factors among PLHIV. There are some important new findings in this review. First, our review provides both prevalence and incidence rates as well as risk factors for suicide ideation, suicide attempt, and death by suicide among PLHIV. Second, the current study includes findings from diverse populations of patients with HIV from 1988–2021, while previous reviews mostly focused on certain risk populations. Third, our study provides a group association and risk factors for suicidal ideation, suicide attempts, and death due to suicide. Therefore, we believe our findings suggest definite trends and factors that could prevent suicidal behavior among PLHIV, which future studies should examine further.

The limitations of this study were the lack of information regarding ethnic groups, cultural backgrounds, and religious perspectives of suicidal behavior and risk factors among PLHIV. Future studies should focus on these factors prospectively. Also, this large number of studies contained different type of confounding factors and it is difficult to control all confounding one time, however it will not influence to review findings because we would provide overview of suicidal behaviors only. Still did not make any causal relationship furthermore future study designed how to manage confounding such an incident if suicide actions. Also study quality is deferent to each study, not ranked study quality in terms of sample size, biases, etc. same as different scales/ measurement tools were used which also affects consistency in studies can consider some limitations.

## Conclusion

This scoping review presents a global view of suicidal behavior in 49 countries and included 193 primary research studies. We found that the Americas, Europe, and some Asia countries have the highest rates of suicidal behavior also after free access of antiviral therapy and post-HAART era, there has been an increasing trend in suicidal behavior. Depression, low quality of life, low social support, substance use, and drug abuse are the most common risk factors for suicidal behavior. Our study lacks information on ethnicity, cultural background, and religious perspectives of PLHIV, and those need to be considered in future studies. This review provides an overview of suicidal behavior and risk factors for future healthcare development plans and prevention of suicide in PLHIV.

## Clinical applications

This study will provide data on global suicidal ideation, suicide attempts, and completed suicide as well as the epidemiology and risk factors associated with completed suicides among people living with HIV. The findings of this review can be used as scientific evidence in the design of protocols and clinical practice guidelines intended to manage the wellbeing of PLHIV worldwide. It is also a reference for future researchers who plan to examine suicidal behavior and the risk factors among diverse populations. This study has practical implications for the management of people with HIV and preventing suicide at the global level. Given the high prevalence of suicide in high-risk populations such as people with HIV and the challenges related to preventing suicide, our study findings could support suicide prevention efforts by presenting the prevalence and incidence rates for suicide, as well as the associated risk factors among PLHIV.

## Supporting information

**S1 Table. Suicide ideation rate among people living with HIV.**
(DOCX)

**S2 Table. Suicide attempt rate among people living with HIV.**
(DOCX)

**S3 Table. Death due to suicide rate among people living with HIV.**
(DOCX)

**S1 File. SRSearchForm.**
(DOCX)

**S1 Checklist.**
(DOCX)

## Acknowledgments

The authors thank all the other authors of the included studies.

## Author Contributions

**Writing – original draft:** Yi-Tseng Tsai, Sriyani Padmalatha K. M., Han-Chang Ku, Yi-Lin Wu, Nai-Ying Ko.

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
