## [Decision Letter · Decision Letter 0]

11 Feb 2022

PONE-D-21-32836Global overview of suicidal
behavior and associated risk factors among people living with human immunodeficiency
virus: a scoping reviewPLOS ONE

Dear Dr. Ko

Thank you for submitting your manuscript to PLOS ONE. After careful consideration, we
feel that it has merit but does not fully meet PLOS ONE’s publication criteria as it
currently stands. Therefore, we invite you to submit a revised version of the
manuscript that addresses the points raised during the review process.

Please submit your revised manuscript by Mar 28 2022 11:59PM. If you will need more
time than this to complete your revisions, please reply to this message or contact
the journal office at plosone@plos.org. When
you're ready to submit your revision, log on to https://www.editorialmanager.com/pone/ and select the 'Submissions
Needing Revision' folder to locate your manuscript file.

Please include the following items when submitting your revised
manuscript:A rebuttal letter that responds to each point raised by the academic
editor and reviewer(s). You should upload this letter as a separate file
labeled 'Response to Reviewers'.A marked-up copy of your manuscript that highlights changes made to the
original version. You should upload this as a separate file labeled
'Revised Manuscript with Track Changes'.An unmarked version of your revised paper without tracked changes. You
should upload this as a separate file labeled 'Manuscript'.

If you would like to make changes to your financial disclosure, please include your
updated statement in your cover letter. Guidelines for resubmitting your figure
files are available below the reviewer comments at the end of this letter.

We look forward to receiving your revised manuscript.

Kind regards,

Saeed Ahmed, MD

Academic Editor

PLOS ONE

Journal Requirements:

5. We note that figure 2.1 in your submission contain [map/satellite] images which
may be copyrighted. All PLOS content is published under the Creative Commons
Attribution License (CC BY 4.0), which means that the manuscript, images, and
Supporting Information files will be freely available online, and any third party is
permitted to access, download, copy, distribute, and use these materials in any way,
even commercially, with proper attribution. For these reasons, we cannot publish
previously copyrighted maps or satellite images created using proprietary data, such
as Google software (Google Maps, Street View, and Earth). For more information, see
our copyright guidelines: http://journals.plos.org/plosone/s/licenses-and-copyright.

   1. You may seek permission from the original copyright holder of figure 2.1 to
publish the content specifically under the CC BY 4.0 license.  

Maps at the CIA (public domain): https://www.cia.gov/library/publications/the-world-
factbook/index.html and https://www.cia.gov/library/publications/cia-maps-publications/index.html

Reviewer's Responses to Questions

**Comments to the Author**

1. Is the manuscript technically sound, and do the data support the conclusions?

Reviewer #1: Partly

Reviewer #2: Yes

Reviewer #3: Partly

Reviewer #4: Yes

2. Has the statistical analysis been performed
appropriately and rigorously? 

Reviewer #1: Yes

Reviewer #2: Yes

Reviewer #3: Yes

Reviewer #4: N/A

3. Have the authors made all data underlying the
findings in their manuscript fully available?

Reviewer #1: Yes

Reviewer #2: Yes

Reviewer #3: Yes

Reviewer #4: Yes

4. Is the manuscript presented in an intelligible
fashion and written in standard English?

Reviewer #1: No

Reviewer #2: Yes

Reviewer #3: No

Reviewer #4: No

5. Review Comments to the Author

Reviewer #1: This systematic review reviewing global literature to document suicidal
behavior and risk factors among people living with HIV from Jan 1 1988 to July 8
2021. The purpose is to help prevent suicidal risk among PLHIV. It is an original
literature review. With such a wide ranging study, it can be difficult to find
consistent findings or learning points especially with various confounding factors
in various studies which are difficult to adjust for. A number of risk factors and
suicidal behavioral patterns were observed among these papers. The selection of
publications to review was extensive which could lead various inconsistent
conclusions in addition to few consistent ones including psychiatric conditions, low
social support, discrimination. As per authors, previous primary research had shown
poor social support, HIV stigma, mental disorders and associated co-morbidities as
risk factors.

Clearly, the authors have worked hard and gone through a lot of papers for this
manuscript. It was quite interesting and informative to read. I have a few comments/
questions about it that I believe could help make it more suitable for publication.
This includes minor grammatical errors to major questions.

1) Authors have mentioned several times (e.g.- Line 32, line 204 and also in
conclusion) that suicide rates have increased from the pre HAART era to Post HAART
recently. This seemed unlikely to me and looked into citations you have made.
Citation 5 says suicide decreased to 0. And in citation 63, it says suicide
proportion has increased as aids related cause of death has decreased. Citation 4
mentions "Suicide rates decreased significantly with the introduction of HAART".
Please correct me with evidence that completed suicides have increased since HAART
was brought in.

If you are stating that suicidal rates have in fact increased in this time, could you
mention why that is? Which risk factors have worsened? when you say suicide rates
have increased, do you mean suicides are causing higher proportion of deaths because
people are living longer due to HAART and not dying due to AIDS as much? If so, that
needs clarifying.

2) Line 28- 800k died every year - in which years or is that an average since 1988?
Please clarify

3) Line 50-53: Review sentence grammar

4) Line 47: Relation between 3 suicide behaviors- It would be interesting if you
could specify certain risk factors that are more associated with either one of these
3 suicidal behaviors separately or other different relationships between the 3. Just
a suggestion.

5) Line 165- Pre* HAART not Per*

6) Line 225- Some risk factors are confusing such as 'religion', 'gender', 'age'.
Could you specify what the papers say about how these factors affect risk? Which
ages, genders, etc?

7) Discussion- Line 253-255- "Suicide prevalence highest in Americas/ Europe from
1997 to 2020" but then you list countries that has Australia, Thailand, South Africa
as top prevalence in the next few lines.

8) Discussion- Line 259-Line 261: You mentioned reasons as liberal laws on
doctor-assistant suicide, gun control or economic/ social pressures: Any citations
for that?

9) Discussion- Line 268- Line 270- "85% of suicides occur in Africa and middle/low
income countries": Citation?

10) You could mention higher risk of suicide in people with HIV compared to general
population to emphasize suicidal risks in PLHIV (as seen in citations you have
provided)

11) Discussion- Line 274- In Asian region, most common suicide behavior is completed
suicide. So increased completed suicides compared to other regions (that goes
against previous report that suicidal behaviors highest in Americas/ Europe) or
completed suicides are higher than suicidal ideation/ attempts (That's not
possible)

12) Discussion- Line 278 - "Most common death factor?"; "Depression and its common
cause." Please correct that whole sentence to be clear grammatically.

13) In Discussion- It would help to mention something the reader should take from
this study that could help clinically such as importance of social supports, etc in
HIV treatment

14) Conclusion- "We found that Americas, Europe and Asia have the highest rates of
suicidal behavior" - Compared to what? Africa and Australia? Better if conclusion is
more specific. Please change the wording to explain what you mean.

15) How did your results compare to what we knew from previous research studies-
Confirmed some? Expanded any additional information we didn't know before? Add them
in conclusion as important findings from this study.

16) The figures and tables are well done. Would help to have a table with consistent/
inconsistent risk factors

17) There are several limitations you can add at the end.

1) Different types of studies can have different qualities of study. Not evaluating
which studies are better in terms of sample size, biases, etc. is a limitation.

2) Given different time periods, cultures etc, in regional studies review, it is
difficult to make generalized conclusions for global factors and patterns. Recent
studies more valuable than pre-HAART studies.

3) Different scales/ measurement tools were used which also affects consistency in
studies.

18) Many grammatical errors and spelling errors noted in the manuscript. It would be
helpful to review it once more.

Reviewer #2: some of the sentences are repetitive specially definitions of different
suicidal behaviors including ideations, and attempts and can be revised. Good
description of consistent and inconsistent risk factors.

Reviewer #3: The submitted manuscript discusses an important topic of suicide in
patients with HIV. However in its current form, the article suffers from some
deficiencies which need to be addressed. A lot of focus has been placed on tabular
display of data rather than the written sections. The introduction section
especially has been presented in a very matter of fact fashion and does not evoke
much curiosity in the reader. The authors need to cite more articles to drive home
the point that suicide in patients with HIV is a matter of public health concern. It
would be helpful to tie in Hypersexual behaviors, sexual addiction and other
psychiatric disorders to HIV.

I refer the authors to the following article which discussed hypersexuality and
sexual addiction to risk of getting STDs. Kindly review and cite the article as
appropriate.

https://www.researchgate.net/profile/Ashish-Sarangi-2/publication/320741412_Hypersexual_Disorder-A_Case_Report_and_Analysis_A_R_T_I_C_L_E_I_N_F_O/links/59f916ea0f7e9b553ec0c8ec/Hypersexual-Disorder-A-Case-Report-and-Analysis-A-R-T-I-C-L-E-I-N-F-O.pdf

Although not the primary aim, it would be helpful to include a few lines in the
discussion section making suggestions on how to screen patients at risk of suicide
and refer for appropriate interventions.

It will also be helpful to discuss the main methods used for suicide if this data is
available for example hanging e.t.c

Please condense the tables as there is too much data in the tables for meaningful
review.

I will be happy to accept a revised manuscript for consideration as I believe it has
potential.

Reviewer #4: Thank you for the opportunity to review the article.

The review is an excellent and balanced overview that includes 193 observational
studies, a large sample size from 49 countries, both clinic and hospital data from
multiple databases, encompasses diverse populations, 23 years of data( 2/3 of papers
were recent), gives an interesting perspective on historical pre HAART suicide rate
compared to post HAART suicide rate in HIV patient that ironically showed an
increasing trend of suicidal behavior post-HAART therapy which is essential to be
addressed and is often overlooked. Furthermore, they have highlighted risk factors
of suicidal ideation and complete suicide in detail in different countries(included
physical, psychological, social, demographic risk factors), also used standard
scales as their guide, talked about methods of committing suicide in different
countries, information on prevalence, association with certain coexisting conditions
like substance disorder, mental illness, etc. They have registered in INPLASY, has
Prisma flow diagrams, appropriate methodology for review and bias, did JBI for ROB
assessment, and have good review article tables and flow charts.

However, I have few suggestions that could be added to the article.

In my opinion, a key feature that could be included in the article is, using all
patient sample data from various articles divided into baseline characteristics such
as age, sex, ethnicity, family history data, cultural and social aspects, important
comorbid conditions like substance use disorder, stigma, mental illness, and side
effects of any HAART therapy causing suicidal ideation. Then, Subsequently analyzed
the total patient sample to derive any associations and protective factors. Please
elaborate more on the third point mentioned in the discussion part and, if able to,
try to describe any methods that can tackle this situation of suicide in post HAART
HIV patients with a positive angle. Language is unclear at certain parts of the
papers, making it difficult to follow. Therefore, I advise the authors to revise
with Minor phrasing and grammatical issues to improve the flow and readability of
the text.

6. PLOS authors have the option to publish the peer
review history of their article (what does this mean?). If published, this will
include your full peer review and any attached files.

If you choose “no”, your identity will remain anonymous but your review may still be
made public.

**Do you want your identity to be public for this peer review?** For
information about this choice, including consent withdrawal, please see our
Privacy Policy.

Reviewer #1: No

Reviewer #2: **Yes: **Meenal Pathak

Reviewer #3: No

Reviewer #4: **Yes: **Wasey Ali Yadullahi Mir

---

## [Author Response · Author response to Decision Letter 0]

28 Apr 2022

Dear Professors, Saeed Ahmed,

We highly appreciate your constructive comments and valuable suggestions to our
manuscript. We have edited this manuscript according to your journal additional
requirements and have revised our manuscript highlight with red colour according to
all reviewers’ suggestions, and explain the changes as noted below:

Responses to Reviewers: 

Reviewer #1: 

1. This systematic review reviewing global literature to document suicidal behavior
and risk factors among people living with HIV from Jan 1 1988 to July 8 2021. The
purpose is to help prevent suicidal risk among PLHIV. It is an original literature
review. With such a wide-ranging study, it can be difficult to find consistent
findings or learning points especially with various confounding factors in various
studies which are difficult to adjust for A number of risk factors and suicidal
behavioral patterns were observed among these papers. The selection of publications
to review was extensive which could lead various inconsistent conclusions in
addition to few consistent ones including psychiatric conditions, low social
support, discrimination. As per authors, previous primary research had shown poor
social support, HIV stigma, mental disorders and associated co-morbidities as risk
factors.

Clearly, the authors have worked hard and gone through a lot of papers for this
manuscript. It was quite interesting and informative to read. I have a few comments/
questions about it that I believe could help make it more suitable for publication.
This includes minor grammatical errors to major questions.

(1) Authors have mentioned several times (e.g.- Line 32, line 204 and also in
conclusion) that suicide rates have increased from the pre-HAART era to Post HAART
recently. This seemed unlikely to me and looked into citations you have made.
Citation 5 says suicide decreased to 0. And in citation 63, it says suicide
proportion has increased as aids related cause of death has decreased. Citation 4
mentions "Suicide rates decreased significantly with the introduction of HAART".
Please correct me with evidence that completed suicides have increased since HAART
was brought in.

Response: Thank you for the comments. We have changed and added new citation as “The
rate of suicide deaths in People living with HIV (PLHIV) is 100-fold higher than the
rate that has been reported in the general population[3]. Prevalence estimates of
suicidal ideation, attempts, and plans among people living with HIV/AIDS were more
common and serious than those in the general population[4]. Suicide attempt rates
among PLHIV with mental disorders and psychiatric treatment have continued to
increase from the pre-highly active antiretroviral therapy (Pre-HAART) era
(1988–1995) to the HAART era (1996–2008) from 27.8% to 35.1%, respectively[5].
Please refer the introduction section and line no 56 to 63 and Reference no 3, 4, 5. 

(2) If you are stating that suicidal rates have in fact increased in this time, could
you mention why that is? Which risk factors have worsened? when you say suicide
rates have increased, do you mean suicides are causing higher proportion of deaths
because people are living longer due to HAART and not dying due to AIDS as much? If
so, that needs clarifying.

Response: Thank you for the suggestions and have added as “Suicide attempt rates
among PLHIV with mental disorders and psychiatric treatment have continued to
increase from the pre-highly active antiretroviral therapy (Pre-HAART) era
(1988–1995) to the HAART era (1996–2008) from 27.8% to 35.1%, respectively[5].
Please refer the introduction section and line no 60 to 63 and Reference no 5. 

(3) Line 28- 800k died every year - in which years or is that an average since 1988?
Please clarify.

Response: Thank you for comments. We changed as “approximately 700,000 people died
worldwide due to suicide every year [1].” Please refer introduction section and line
no 50 to 52.

(4) Line 50-53: Review sentence grammar.

Response: Thank you for the comment. We corrected grammar in this manuscript using
scribendi.com for this time.

(5) Line 47: Relation between 3 suicide behaviors- It would be interesting if you
could specify certain risk factors that are more associated with either one of these
3 suicidal behaviors separately or other different relationships between the 3. Just
a suggestion.

Response: Thank you for the suggestions and we simply changed as per your suggestion
like this “In the current study provided insights into the relationships among
HARRT, depression, and suicidal status in PLHIV and evidence that depression played
a mediating role in the association between suicide ideation and attempt. However,
the relationship between these three-suicide behavior is unclear; for example,
relationship between HARRT, and death by suicide or depression, and suicide
attempts, therefore, this study will be a better feasibility to understanding
relationship between these three-suicide behavior.” Please refer introduction in
line number 75 to 81. 

(6) Line 165- Pre* HAART not Per*.

Response: Thank you for the correction. We changed as “Pre” please refer result
section and line no 204. 

(7) Line 225- Some risk factors are confusing such as 'religion', 'gender', 'age'.
Could you specify what the papers say about how these factors affect risk? Which
ages, genders, etc?

Response: Thank you for your suggestions. We have changed as “gender (male [4, 34,
35, 84, 93, 114, 122], female [42, 95, 108]), age (young age [3, 60, 78, 80, 140,
160, 167], middle age [10, 19, 83, 144, 182], older age [11]), and religion [144,
151, 197].” Please refer result section and line 252 to 253 and 267.

(8) Discussion- Line 253-255- "Suicide prevalence highest in Americas/ Europe from
1997 to 2020" but then you list countries that has Australia, Thailand, South Africa
as top prevalence in the next few lines.

Response: Thank you for the comments. We changed as “United States, United Kingdom,
Australia, and Russia for suicidal ideation, suicide attempts rate was highest in
the United States, Australia, and Spain, and death due to suicide rate was highest
in Denmark, and Thailand among PLHIV from 2000 to 2020.” Please see the discussion
section line 297 t0 299. 

(9) Discussion- Line 259-Line 261: You mentioned reasons as liberal laws on
doctor-assistant suicide, gun control or economic/ social pressures: Any citations
for that?

Response: Thank you for the suggestions. We added these citations [27, 29, 209,
210].

(10) Discussion- Line 268- Line 270- "85% of suicides occur in Africa and
middle/low-income countries": Citation?

Response: Thank you for the suggestions. We simply removed that sentence because it
has some typo missing and error information. 

(11) You could mention higher risk of suicide in people with HIV compared to general
population to emphasize suicidal risks in PLHIV (as seen in citations you have
provided).

Response: Thank you for the suggestions. We have changed our introductions as “The
rate of suicide deaths in People living with HIV (PLHIV) is 100-fold higher than the
rate that has been reported in the general population[3].” and added one reference
(Ref: No 3). Also, in the discussion we started as “About 40 million people of the
global population are currently living with HIV/AIDS. The era of HAART treatment has
brought significant improvements in patient longevity and quality of life; however,
PLHIV experience a heavy burden of psychosocial conditions that are frequently
undiagnosed and untreated. The pooled incidence of suicide completion among PLHIV
globally was 10.2 per 1000 population, translating to a 100-fold greater suicide
completion rate compared with the global population rate of 0.09/1000 population for
2019[3, 210]. With two citations. Please refer line 56 to 57 in introduction and
line 276 to 282 in discussion.

(12) Discussion- Line 274- In Asian region, most common suicide behavior is completed
suicide. So increased completed suicides compared to other regions (that goes
against previous report that suicidal behaviors highest in Americas/ Europe) or
completed suicides are higher than suicidal ideation/ attempts (That's not
possible).

Response: Thank you for the comments. We changed as “According to our finding in when
we considered about South-East Asian Region, most common suicide behaviour is death
due to suicide, compared with suicide attempts and suicide ideation. Because of
educational status of family and social pressure also the social discrimination and
stigma are more common in Asian countries than elsewhere in the world [213, 214].
Please refer line 323 to 327 in discussion section. Send for English editing as
well.

(13) Discussion- Line 278 - "Most common death factor?"; "Depression and its common
cause." Please correct that whole sentence to be clear grammatically.

Response: Thank you for the suggestion. We change as “Depression and suicidal
thinking occur frequently alongside HIV/AIDS, triggering profound detrimental
impacts on quality of life, treatment adherence, disease progression, and
mortality[182, 216]. According to this scoping review, 85 articles dealing with
depression, the most common death-related factor for PLHIV is suicide ideation, and
their attempted suicide behavior risk is due to depression which is its common
cause.” Please refer line328 to 333 in discussion.

(14) In Discussion- It would help to mention something the reader should take from
this study that could help clinically such as importance of social supports, etc in
HIV treatment.

Response: Thank you for the suggestions: we added subtopic of clinical applications
as “The findings of this review can be used as scientific evidence in the design of
protocols and clinical practice guidelines intended to manage the wellbeing of PLHIV
worldwide. It is also a reference for future researchers who plan to examine
suicidal behavior and the risk factors among diverse populations. This study has
practical implications for the management of people with HIV and preventing suicide
at the global level. Given the high prevalence of suicide in high-risk populations
such as people with HIV and the challenges related to preventing suicide, our study
findings could support suicide prevention efforts by presenting the prevalence and
incidence rates for suicide, as well as the associated risk factors among PLHIV.”
Please prefer line no 373 to 382 in clinical application section.

(15) Conclusion- "We found that Americas, Europe and Asia have the highest rates of
suicidal behavior" - Compared to what? Africa and Australia? Better if conclusion is
more specific. Please change the wording to explain what you mean.

Response: Thank you for the suggestions. We change as “We found that the Americas,
Europe, and some Asia countries have the highest rates of suicidal behavior also
after free access of antiviral therapy and post-HAART era, there has been an
increasing trend in suicidal behavior.” Please prefer line No 361 to 364 in
conclusion.

(16) How did your results compare to what we knew from previous research studies-
Confirmed some? Expanded any additional information we didn't know before? Add them
in conclusion as important findings from this study.

Response: Thank you for the suggestions. We added one section as a clinical
application as “The findings of this review can be used as scientific evidence in
the design of protocols and clinical practice guidelines intended to manage the
wellbeing of PLHIV worldwide. It is also a reference for future researchers who plan
to examine suicidal behavior and the risk factors among diverse populations. This
study has practical implications for the management of people with HIV and
preventing suicide at the global level. Given the high prevalence of suicide in
high-risk populations such as people with HIV and the challenges related to
preventing suicide, our study findings could support suicide prevention efforts by
presenting the prevalence and incidence rates for suicide, as well as the associated
risk factors among PLHIV.” Please refer line No: 373 to 382 in clinical application
section. 

(17) The figures and tables are well done. Would help to have a table with
consistent/ inconsistent risk factors.

Response: Thank you for the comments. We have added some new data to table 01 and
table 02 contained data of consistent/ inconsistent risk factors. 

(18) There are several limitations you can add at the end.

○1Different types of studies can have different qualities of study. Not evaluating
which studies are better in terms of sample size, biases, etc. is a limitation.

○2Given different time periods, cultures etc, in regional studies review, it is
difficult to make generalized conclusions for global factors and patterns. Recent
studies more valuable than pre-HAART studies.

○3Different scales/ measurement tools were used which also affects consistency in
studies.

Response: Thank you for the suggestions. We changed our limitations as “Also, this
large number of studies contained different type of confounding factors and it is
difficult to control all confounding one time, however it will not influence to
review findings because we would provide overview of suicidal behaviors only. Still
did not make any causal relationship furthermore future study designed how to manage
confounding such an incident if suicide actions. Also study quality is deferent to
each study, not ranked study quality in terms of sample size, biases, etc. same as
different scales/ measurement tools were used which also affects consistency in
studies can consider some limitations”. Please refer line No: 348 to 355 in
limitation section. 

(19) Many grammatical errors and spelling errors noted in the manuscript. It would be
helpful to review it once more.

Response: Thank you for the suggestions. We have sent this manuscript for English
editing by scribendi.com for this time. 

Reviewer #2: 

Some of the sentences are repetitive specially definitions of different suicidal
behaviours including ideations and attempts and can be revised. Good description of
consistent and inconsistent risk factors.

Response: Thank you for your great feedback. 

Reviewer #3:

The submitted manuscript discusses an important topic of suicide in patients with
HIV. However, in its current form, the article suffers from some deficiencies which
need to be addressed. A lot of focus has been placed on tabular display of data
rather than the written sections. The introduction section especially has been
presented in a very matter of fact fashion and does not evoke much curiosity in the
reader. The authors need to cite more articles to drive home the point that suicide
in patients with HIV is a matter of public health concern. It would be helpful to
tie in Hypersexual behaviours, sexual addiction, and other psychiatric disorders to
HIV.

I refer the authors to the following article which discussed hypersexuality and
sexual addiction to risk of getting STDs. Kindly review and cite the article as
appropriate.

https://www.researchgate.net/profile/Ashish-Sarangi
2/publication/320741412_Hypersexual_Disorder-A_Case_Report_and_Analysis_A_R_T_I_C_L_E_I_N_F_O/links/59f916ea0f7e9b553ec0c8ec/Hypersexual-Disorder-A-Case-Report-and-Analysis-A-R-T-I-C-L-E-I-N-F-O.pdf

Although not the primary aim, it would be helpful to include a few lines in the
discussion section making suggestions on how to screen patients at risk of suicide
and refer for appropriate interventions.

It will also be helpful to discuss the main methods used for suicide if this data is
available for example hanging e.t.c

Please condense the tables as there is too much data in the tables for meaningful
review.

I will be happy to accept a revised manuscript for consideration as I believe it has
potential.

Response: Thank you for the comments and suggestions. We cited this article in
discussion and changed some table and figure data again also the rewrite some area
in introduction, results and discussion as per your suggestions. Please refer line
No: 56 to 81, and No: 275 to 355 in limitation section.

Reviewer #4:

Thank you for the opportunity to review the article.

The review is an excellent and balanced overview that includes 193 observational
studies, a large sample size from 49 countries, both clinic and hospital data from
multiple databases, encompasses diverse populations, 23 years of data( 2/3 of papers
were recent), gives an interesting perspective on historical pre HAART suicide rate
compared to post HAART suicide rate in HIV patient that ironically showed an
increasing trend of suicidal behavior post-HAART therapy which is essential to be
addressed and is often overlooked. Furthermore, they have highlighted risk factors
of suicidal ideation and complete suicide in detail in different countries (included
physical, psychological, social, demographic risk factors), also used standard
scales as their guide, talked about methods of committing suicide in different
countries, information on prevalence, association with certain coexisting conditions
like substance disorder, mental illness, etc. They have registered in INPLASY, has
Prisma flow diagrams, appropriate methodology for review and bias, did JBI for ROB
assessment, and have good review article tables and flow charts.

However, I have few suggestions that could be added to the article.

In my opinion, a key feature that could be included in the article is, using all
patient sample data from various articles divided into baseline characteristics such
as age, sex, ethnicity, family history data, cultural and social aspects, important
comorbid conditions like substance use disorder, stigma, mental illness, and side
effects of any HAART therapy causing suicidal ideation. Then, subsequently analysed
the total patient sample to derive any associations and protective factors. Please
elaborate more on the third point mentioned in the discussion part and, if able to,
try to describe any methods that can tackle this situation of suicide in post HAART
HIV patients with a positive angle. Language is unclear at certain parts of the
papers, making it difficult to follow. Therefore, I advise the authors to revise
with Minor phrasing and grammatical issues to improve the flow and readability of
the text.

Response: Thanks for your valuable comments and suggestions. We have rechecked all
the data again and updated table and figure. Also write rewrite some section in
introductions, result and discussions also modified limitations. Added one sections
for clinical applications. Please refer line No: 56 to 81, and No: 275 to 355 in
limitation section.

reviewers(1).docx
---

## [Decision Letter · Decision Letter 1]

23 May 2022

Global overview of suicidal behavior and associated risk factors among people living
with human immunodeficiency virus: A scoping review

PONE-D-21-32836R1

Dear Dr. Ko,

We’re pleased to inform you that your manuscript has been judged scientifically
suitable for publication and will be formally accepted for publication once it meets
all outstanding technical requirements.

Kind regards,

Saeed Ahmed, MD

Academic Editor

PLOS ONE

**Comments to the Author**

1. If the authors have adequately addressed your comments raised in a previous round
of review and you feel that this manuscript is now acceptable for publication, you
may indicate that here to bypass the “Comments to the Author” section, enter your
conflict of interest statement in the “Confidential to Editor” section, and submit
your "Accept" recommendation.

Reviewer #2: All comments have been addressed

Reviewer #3: All comments have been addressed

Reviewer #4: All comments have been addressed

2. Is the manuscript technically sound, and do the data
support the conclusions?

Reviewer #2: Yes

Reviewer #3: Yes

Reviewer #4: Yes

3. Has the statistical analysis been performed
appropriately and rigorously? 

Reviewer #2: Yes

Reviewer #3: Yes

Reviewer #4: No

4. Have the authors made all data underlying the
findings in their manuscript fully available?

Reviewer #2: Yes

Reviewer #3: Yes

Reviewer #4: Yes

5. Is the manuscript presented in an intelligible
fashion and written in standard English?

Reviewer #2: Yes

Reviewer #3: Yes

Reviewer #4: Yes

6. Review Comments to the Author

Reviewer #2: Authors seem to have incorporated the comments and feedback in their
update. It is an important topic, and the data is analyzed extensively.

Reviewer #3: The reviewer comments have been adequately addressed. Happy to accept
for publication as it will be a good addition.

Reviewer #4: (No Response)

---

## [Editor Report · Acceptance letter]

17 Aug 2022

PONE-D-21-32836R1 

Global overview of suicidal behavior and associated risk factors among people living
with human immunodeficiency virus: a scoping review 

Dear Dr. Ko:

I'm pleased to inform you that your manuscript has been deemed suitable for
publication in PLOS ONE. Congratulations! Your manuscript is now with our production
department. 

Kind regards, 

on behalf of

Dr. Saeed Ahmed 

Academic Editor

PLOS ONE